# Iron Oxides Nanoparticles as Components of Ferroptosis-Inducing Systems: Screening of Potential Candidates

Artur Dzeranov [1,2,*], Lyubov Bondarenko [1,*], Denis Pankratov [3], Mikhail Prokof'ev [1], Gulzhian Dzhardimalieva [1,4], Sharipa Jorobekova [5], Nataliya Tropskaya [1,2], Ludmila Telegina [6] and Kamila Kydralieva [1]

1   Moscow Aviation Institute (National Research University), Moscow 125993, Russia
2   Sklifosofskiy Research Institute of Emergency Medicine, Moscow 129010, Russia
3   Department of Chemistry, Lomonosov Moscow State University, Moscow 119991, Russia
4   Federal Research Center of Problems of Chemical Physics and Medicinal Chemistry, Russian Academy of Sciences, Chernogolovka 142432, Moscow region, Russia
5   Institute of Chemistry and Phytotechnologies, National Academy of Sciences, Bishkek 720071, Kyrgyzstan
6   Russian Institute for Scientific and Technical Information, Moscow 125190, Russia
*   Correspondence: arturdzeranov99@gmail.com (A.D.); l.s.bondarenko92@gmail.com (L.B.)

**Abstract:** This study presents an analysis of a set of iron oxides nanoparticles (NPs) ($\gamma$-Fe$_2$O$_3$, $\alpha$-FeOOH, $\delta$-FeOOH, 5Fe$_2$O$_3$·9H$_2$O, and Fe$_3$O$_4$) as potential candidates for ferroptosis therapy in terms of a phase state, magnetic characteristics, and the release of Fe$^{2+}$/Fe$^{3+}$ as ROS mediators. Due to the values of saturation magnetization for Fe$_3$O$_4$ (31.6 emu/g) and $\gamma$-Fe$_2$O$_3$ (33.8 emu/g), as well as the surface area of these particles (130 and 123 m$^2$/g), it is possible to consider them as promising magnetically controlled carriers that can function with various ligands. The evaluation of the release of Fe$^{2+}$/Fe$^{3+}$ ions as catalysts for the Fenton reaction showed that the concentration of the released ions increases within first 3 h after suspension and decreases within 24 h, which probably indicates desorption and adsorption of ions from/onto the surface of nanoparticles regardless their nature. The concentration of ions released by all nanoparticles, except $\alpha$-FeOOH-Fe$^{2+}$, reached 9.1 mg/L for Fe$^{3+}$ to 1.7 mg/L for Fe$^{2+}$, which makes them preferable for controlling the catalysis of the Fenton reaction. In contrast, a high concentration of iron ions to 90 mg/L for Fe$^{3+}$ and 316 mg/L for Fe$^{2+}$ released from compound $\alpha$-FeOOH-Fe$^{2+}$ allows us to utilize this oxide as an aid therapy agent. Results obtained on iron oxide nanoparticles will provide data for the most prospective candidates that are used in ferroptosis-inducing systems.

**Keywords:** ferroptosis-inducing systems; iron oxide nanoparticles; phase state; specific surface; magnetic properties; ferrous and ferric ions release

## 1. Introduction

The iron-based nanoparticles have become widely used in cancer therapy research thanks to the special physical and chemical properties of nanomaterials. Ferroptosis is a new iron-dependent form of programmed cell death, first described in 2012 by Dixon, and it is one of the types of regulated cell deaths caused by strong activation of lipid peroxidation, which depends on the formation of reactive oxygen species (ROS) and the availability of iron in cells [1,2]. Currently, ferroptosis-inducing systems initiate the Fenton reaction and overexpression of H$_2$O$_2$ in a cancer cell [3,4] as well as the subsequent formation of hydroxyl radicals (OH, one of the most toxic types of ROS) complexes [5–7]. In addition, various nanoplatforms based on iron (II, III) [3,8–10] are used, including ferromagnetic nanoparticles (maghemite $\gamma$-Fe$_2$O$_3$ or magnetite Fe$_3$O$_4$) [11,12] and complexes or their nanocomposites [13,14]. To our best knowledge, there have never been works devoted to the comparative evaluation of various iron-containing compounds as ferroptosis catalysts.

One of the factors affecting the Fenton reaction is the type of iron ions inducing the Fenton reaction by $Fe^{3+}$ or the $Fe^{2+}/Fe^{3+}$ pair. The Fenton reaction is catalyzed by $Fe^{2+}$, and the Fenton-mediated reaction is catalyzed by $Fe^{3+}$; both of them result in the production of toxic reactive oxygen species. In 1987, Minotti et al. showed [15] that lipid peroxidation can be induced both by the participation of separate $Fe^{2+}$ or $Fe^{3+}$ through the formation of hydroxyl radicals as a result of the Fenton reaction and with the participation of a pair of $Fe^{2+}/Fe^{3+}$ at once. Thus, He et al. (2020) obtained a lipid peroxidation catalyst containing a pair of $Fe^{2+}/Fe^{3+}$, where $Fe^{2+}$ was reduced from $Fe^{3+}$ by glutathione, an antioxidant enzyme found in high concentrations in cancer cells [16]. The authors reported the antitumor activity of the catalysts prepared in vitro and in vivo experiments against MCF-7/ADR breast cancer cells, MCF-7 breast cancer cells, 4T1 mouse breast cancer cells, and human liver cancer L02 cells without side effects. However, the resulting nanocatalysts cannot be used for targeted delivery due to the lack of magnetic properties, Huo et al., 2017 [12] showed that in the absence of an external magnetic field or any other methods of targeting the drug, only 6.95% of $Fe_3O_4$ with glucose oxidase in mesoporous silica framework accumulated in the tumor in 48 h. It is important to mention that there are no papers devoted to the comparative analysis of various iron-containing preparations that could catalyze the Fenton reaction. However, the authors of most publications related to the topic do not make a direct correlation between the concentration of released iron ions, the concentration of hydroxyl radicals, and the percentage of cell death. The use of iron oxide nanoparticles, in particular magnetite $Fe_3O_4$ and maghemite $\gamma$-$Fe_2O_3$, is the most widely used since (1) magnetite $Fe_3O_4$ nanoparticles contain both $Fe^{2+}$ and $Fe^{3+}$; (2) $\gamma$-$Fe_2O_3$ nanoparticles are the source of $Fe^{3+}$ ions, which can be reduced to $Fe^{2+}$ with ascorbic acid; and (3) $Fe_3O_4$ magnetite nanoparticles, like $\gamma$-$Fe_2O_3$ nanoparticles, have magnetic properties that provide the means to control them with the application of an external magnetic field.

An open issue in the design of iron preparations is the issue with the capability of the controlled induction of ROS in sufficient quantities in Fenton reactions to kill a cancer cell. It needs a high drug load, high yield, and controlled release of $Fe^{2+}$ and $Fe^{3+}$ ions; targeted delivery of the drug to the tumor; surface modification to control the "fate" in vivo and ensure stability; and effective detection using imaging techniques.

This study aimed to evaluate various types of bare iron oxide nanoparticles to be a core and iron ion source of nanosystems inducing reactive oxygen species (ROS) in terms of their phase state, magnetic properties, kinetics, and release concentration of ferrous and ferric ions. Iron oxide nanoparticles, nanoparticles of magnetite $Fe_3O_4$, and maghemite $\gamma$-$Fe_2O_3$ were chosen as samples due to superparamagnetic properties [17,18], feroxyhyte $\delta$-FeOOH, due to a large specific surface [19] and a significant number of surface –OH groups [20], and a good water solubility as well, ferrihydrite $5Fe_2O_3 \cdot 9H_2O$ and goethite $\alpha$-FeOOH were chosen due to high reactivity with respect to dissolved organic substances [21–23] and a highly hydrated structure.

The use of different iron nanoparticles in order to search for effective iron preparations usually fails to achieve the desired properties using one universal preparation. On the contrary, the use of a set of iron nanoparticle preparations with subsequent analysis of the phase composition, magnetic characteristics, and surface characteristics determines the effectiveness of their use.

## 2. Materials and Methods

### 2.1. Chemicals

$FeCl_3 \cdot 6H_2O$ (Brom, Russia), $FeSO_4 \cdot 7H_2O$ (Chemicals, Russia), 10% NaOH (Kaustik, Russia), 30% $H_2O_2$ (Lega, Russia), $(NH_4)_2Fe(SO_4)_2 \cdot 6H_2O$ (MZKhR, Russia), $NaBH_4$ (Chemical Line, Russia). All reagents were chemically pure.

### 2.2. Preparation of $Fe_3O_4$

Magnetite $Fe_3O_4$ nanoparticles were prepared by the coprecipitation of aqueous solutions of iron (3+) and (2+) salts in the presence of alkali according to the Elmor method [24].

For that, 11.5 g of $FeCl_3 \cdot 6H_2O$ and 6 g of $FeSO_4 \cdot 7H_2O$ were dissolved in 0.5 L of deionized $H_2O$. The solution was heated to 60 °C. While stirring on a mechanical stirrer (1000 rpm), 10% NaOH was added to pH 10. The resulting precipitate of magnetite was separated from the solution using a magnet (Nd; 0.3 T) and applying it to the wall of the reaction flask. To remove $Na^+$, $Cl^-$, $SO_4^{2-}$, and $OH^-$, the precipitate was washed 3 times with deionized water to pH~8. The sample was dried in a convector drier at 50 °C.

### 2.3. Preparation of γ-Fe₂O₃

A total of 10 g of maghemite nanoparticles was obtained by air-oxidation of a sample of magnetite nanoparticles in 20 mL 30% $H_2O_2$ in 5 days. The suspension was centrifuged within 10 min at 3000 rpm. The formulated nanoparticles were twice washed with deionized water and centrifuged (10 min, 3000 rpm). The sample was dried in the thermostat (105 °C, 24 h).

### 2.4. Preparation of δ-FeOOH

One-step synthesis of δ-FeOOH was performed according to the method used by Pinto et al. [25]. A total of 400 mL of a solution containing 11 g of $(NH_4)_2Fe(SO_4)_2 \cdot 6H_2O$ was stirred on a mechanical stirrer (500 rpm, 10 min), and 150 mL of 10% NaOH was gradually added. A green suspension was formed. Thereafter, 5 mL of 30% $H_2O_2$ was immediately added, and a reddish brown precipitate was formed subsequently. The resulting solid precipitate was separated by centrifugation (10 min, 3000 rpm). The precipitate was washed with 800 mL of deionized water to pH ~8 and centrifuged (10 min, 3000 rpm). The sample was dried in a convector drier at 50 °C for 24 h.

### 2.5. Preparation of 5Fe₂O₃·9H₂O

Ferrihydrite NPs ($5Fe_2O_3 \cdot 9H_2O$) were obtained according to [26] as follows: 21.6 g of $FeCl_3 \cdot 6H_2O$ was dissolved in 400 mL of deionized water and then titrated with 1 M NaOH solution to pH~7.0 by continuous stirring (600 rpm) in a magnetic stirrer. The 0.1 M NaOH was adjusted to pH = 7.5 to form dark red solid ferrihydrite. The resulting solid was separated by centrifugation (20 min, 3000 rpm). The precipitate was washed with 400 mL of deionized water to pH = 6 and centrifuged (20 min, 3000 rpm). The sample was dried in a convector drier at 50 °C within 24 h.

### 2.6. Preparation of α-FeOOH-Fe²⁺

Due to the need to dope α-FeOOH by $Fe^{2+}$ and prevent rapid oxidation to ferrihydrite, preparation of alpha-FeOOH-$Fe^{2+}$ was adopted from [27] with modification: synthesis environment was changed from inert on ambient conditions and MeOH on EtOH.

An amount of 15 g $FeSO_4 \cdot 7H_2O$ was dissolved in 600 mL 70% deionized water and 30% ethanol. Using 1 M NaOH, the pH of the solution was adjusted to 6.8. Then a solution of sodium borohydride (3.8 g $NaBH_4$ in 150 mL) was added at a rate of 2 drops/s with constant stirring on a mechanical stirrer for 30 min (pH = 7.4). The solution was centrifuged (10 min, 3000 rpm). A yellow precipitate formed was washed twice with ethanol (200 mL each) and centrifuged (10 min, 3000 rpm). The resulting particles were dried in the thermostat (105 °C, 24 h).

### 2.7. Characterization

X-ray phase analysis was performed on Thermo Fisher Scientific ARL X'TRA diffractometer (Cu $K_\alpha$ radiation ($\lambda$ = 1.54184 Å) in the angle range 2θ = 5–80° at a scanning speed of 5 deg/min and 25 °C. Powder X-ray analysis was performed using Match! software. Mössbauer spectra with a noise/signal ratio of no more than 2% were obtained on an MS1104EM spectrometer using 57Co in an Rh matrix as a source of gamma radiation, with a temperature control accuracy of $\pm 2$ and 0.5 deg. for 295 and 77 K, respectively. The experimental spectra were interpreted using SpectRelax 2.8. The isomer shift was determined relative to α-Fe.

The specific surface area and characteristics of the porous structure of the samples were determined on a Sorptometer-M sorptometer (Katakon, Russia) at liquid nitrogen temperature (77 K). The calculation of the specific surface area and characteristics of the porous structure of the samples based on the adsorption-desorption isotherms was carried out by using the Brunauer–Emmett–Teller (BET) and Barrett–Joyner–Halenda (BJH) methods. Before the start of the tests, the samples were degassed by heating in a He (helium) current, 150 °C, 2 h, flow rate 0.2 L/h in order to remove adsorbed gasses and vapors from the surface.

The magnetic properties of iron NPs were determined by using the VSM M4500 vibrating magnetometer (EG&G PARC, Gaithersburg, Maryland, USA) calibrated using a standard pure nickel sample (weighing 90 mg) with a relative accuracy of $1 \times 10^3$ at room temperature. During the experiment, the value of the magnetic field was changed from 0 to 10 kOe at room temperature, which made it possible to measure the saturation magnetization ($M_s$), a saturation of remanent magnetization ($M_r$), and coercive force ($H_c$) for each sample.

In order to detect iron ions, the powder was suspended in deionized water, then subjected to centrifugation (6000 rpm, 5 min) after 0, 0.5, 1, 3, and 24 h. Appropriate ion detection reagents were added to the supernatant after separation. In order to detect $Fe^{3+}$ ions, 200 µL of potassium thiocyanate KSCN (50% solution) and 200 µL of HCl (18.25% solution) was added to 5 mL of supernatant since the reaction between the $Fe^{3+}$ ions and KSCN proceeded in a strongly acidic medium at pH close to 2. Then the solution was kept for 20 min to reach equilibrium, and the absorption spectrum in the region of 490 nm was measured. In order to detect $Fe^{2+}$ ions, 2 mL phenanthroline $C_{12}H_8N_2 \cdot H_2O$ (2.5% solution) and 600 µL of ammonium acetate buffer solution (250 mL of $NH_4OH$ and 900 mL of glacial acetic acid) were also added to the 5 mL of supernatant. Further, the solution was also kept for 20 min to reach equilibrium (no color change) and examined in the region of 690 nm. The absorbance was detected by UV–Vis-NIR spectrophotometry (Cary UV-Vis-NIR Spectrophotometer, Agilent Technologies)

## 3. Results and Discussion

### 3.1. Microstructure of Iron Nanoparticles

The diffraction patterns of the samples contain narrow and symmetrical peaks indicating the formation of a well-crystallized material for all NPs samples except for ferrihydrite (Figure 1). Ferrihydrite ($5Fe_2O_3 \cdot 9H_2O$) is weakly crystalline due to a highly hydrated structure.

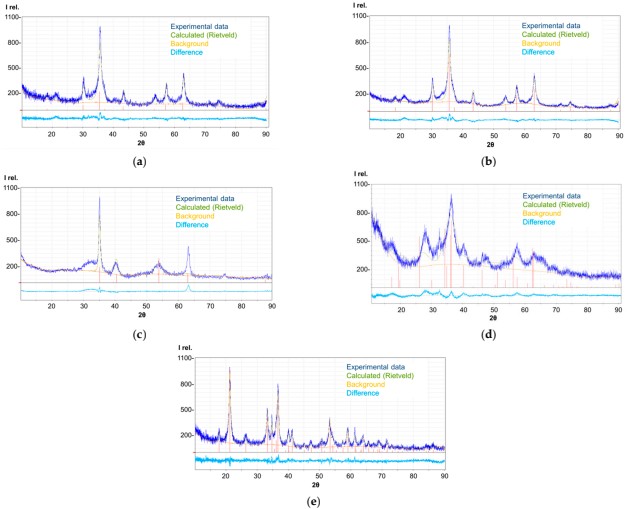

**Figure 1.** Experimental diffraction patterns of samples: (**a**) $Fe_3O_4$, (**b**) $\gamma$-$Fe_2O_3$, (**c**) $\delta$-FeOOH, (**d**) $5Fe_2O_3 \cdot 9H_2O$, (**e**) $\alpha$-FeOOH-$Fe^{2+}$ and the results of their refinement by the Rietveld method.

Quantitative evaluation of XRD data was carried out by the Rietveld method. The main structural parameters of the nanoparticles were determined by using the Crystallography Open Database.

Table 1 presents the lattice parameters of each sample and the Goodness of Fit $(GOF)^2$, which is the $(R_{WP}/R_{exp})^2$ ratio [28]. The $(GoF)^2$ value should approach 1.

**Table 1.** Quantitative phase analysis after Rietveld method.

| Sample | $Fe_3O_4$ | $\gamma$-$Fe_2O_3$ | $\delta$-FeOOH | $5Fe_2O_3 \cdot 9H_2O$ | $\alpha$-FeOOH-$Fe^{2+}$ |
|---|---|---|---|---|---|
| a, A | 8.331 | 8.344 | 2.951 | 5.961 | 4.609 |
| b, A | - | - | - | - | 9.965 |
| c, A | - | - | 4.586 | 8.499 | 3.022 |
| Structure | $Fe_{2.66}O_4$ | $Fe_{2.56}O_4$ | - | - | - |
| $D_{XRD}$, nm | 6.6 ± 0.2 | 6.8 ± 0.2 | 8.1 ± 3.4 | 3.5 ± 0.5 | 12.3 ± 2.8 |
| $(GoF)^2$ | 1.80 | 1.10 | 1.70 | 1.10 | 1.20 |
| Composition determined | $Fe_{2.66}O_4$ | $Fe_{2.56}O_4$ | $\delta$-FeOOH | $5Fe_2O_3 \cdot 9H_2O$ | $\alpha$-FeOOH |

The main phases of all synthesized samples coincide with those predicted, except for $Fe_3O_4$. The lattice parameters of $Fe_3O_4$ and $\gamma$-$Fe_2O_3$ samples obtained in this work are smaller than those known for magnetite (ICDD–PDF 19–629) but larger than for maghemite (ICDD–PDF 39–1346) due to the formation of nonstoichiometric $Fe_{3-\delta}O_4$. The sizes of coherent scattering regions (CSRs) are presented in Table 1. According to Gorski et al., 2010 [29], for magnetite with an ideal $Fe^{2+}$ content (assuming the $Fe_3O_4$ formula), the mineral phase is known as stoichiometric magnetite (x = 0.50). As magnetite becomes oxidized, the $Fe^{2+}/Fe^{3+}$ ratio (Formula (1)) decreases (x < 0.50), with this form denoted as nonstoichiometric or partially oxidized magnetite. The stoichiometry can easily be converted to the following relationship:

$$x = \frac{Fe^{2+}}{Fe^{3+}} = \frac{1-3\delta}{2+2\delta} \tag{1}$$

Finally, the composition of crystalline components of the magnetite and maghemite $Fe_{3-\delta}O_4$ can be assigned as follows: $Fe_{2.66}O_4$ for magnetite and $Fe_{2.56}O_4$ for maghemite (Table 1).

The Mössbauer spectra (MS) of $Fe_3O_4$ and $\gamma$-$Fe_2O_3$ samples are similar to each other and have a strong temperature dependence characteristic of nanosized particles in the superparamagnetic state [30]. At a temperature of 295 K, the spectra are a combination of an intense paramagnetic doublet and a strongly broadened and distorted sextet, described by a polymodal probability function of the distribution of magnetic fields, as shown in Figure 2.

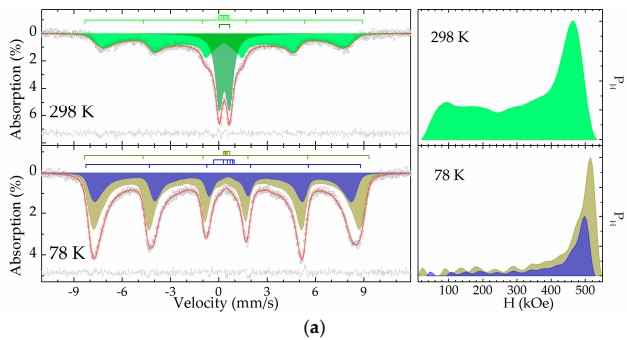

**(a)**

**Figure 2.** *Cont.*

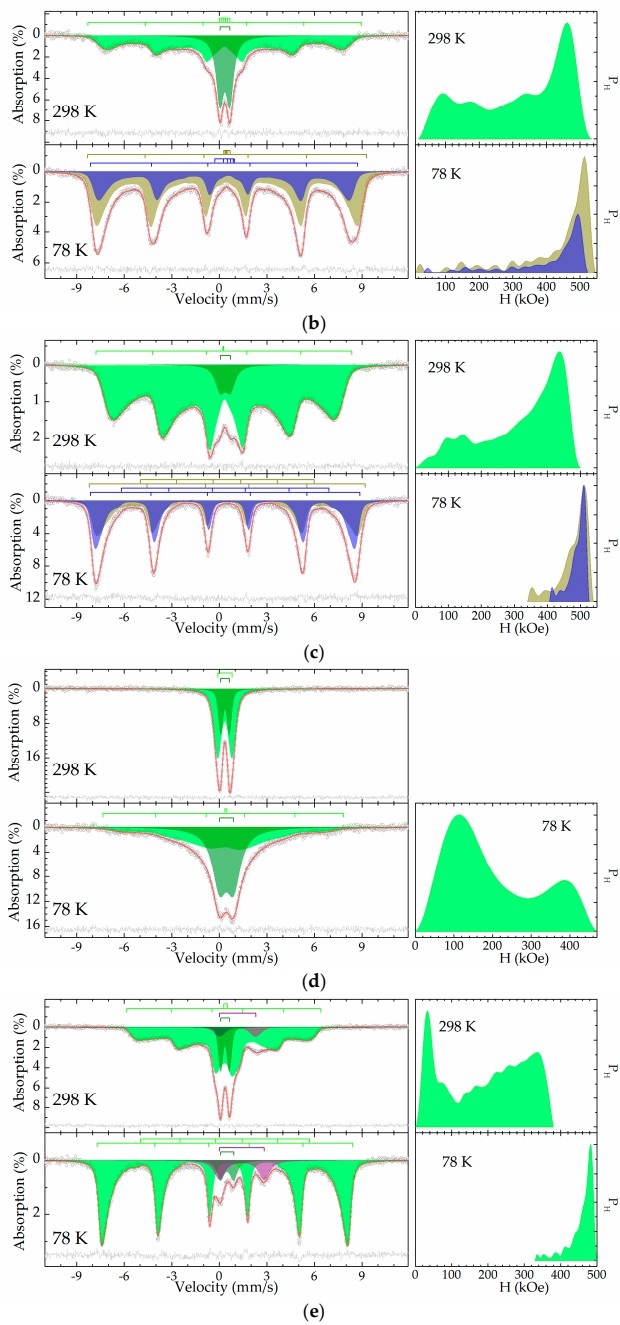

**Figure 2.** Mössbauer spectra at various temperatures, description models, and distribution of the probability function for magnetic fields of samples: (**a**) $Fe_3O_4$, (**b**) $\gamma$-$Fe_2O_3$, (**c**) $\delta$-FeOOH, (**d**) $5Fe_2O_3 \cdot 9H_2O$, (**e**) $\alpha$-FeOOH-$Fe^{2+}$.

At a sample temperature of 78 K, the spectra contain only the resonance lines of the sextet, which are also distorted and asymmetric both in intensity and width, which indicates their composite character. Indeed, the low-temperature spectra of both samples can be satisfactorily described only by a combination of two sextets described by unimodal probability functions for the distribution of magnetic fields with the same profiles and an area ratio of 1/2, which correspond to two iron-containing sub-lattices of the same phase in accordance with the Figure 2. It is obvious that the number of subspectra, the ratio of their areas, and the values of hyperfine parameters allow us to attribute them to the oxidized form of iron oxide with the spinel structure—maghemite [31,32].

Room temperature of the Mössbauer spectra for the $\delta$-FeOOH sample represents continuous absorption over a wide range of velocities, which can be described by a sextet

using a single probability function distribution for a set of Mössbauer hyperfine parameters (isomer shift, quadrupole splitting, and magnetic field) and an additional paramagnetic doublet indicated in Table 2. Cooling the sample to the boiling point of liquid nitrogen leads to the transformation of the MS to a practically symmetrical sextet, which can be described by a combination of two sextets described by unimodal probability distribution functions of magnetic fields with the same profiles and an area ratio of 1/1, which correspond to two iron-containing sublattices of the same phase, as shown in Figure 2. The nature of the temperature dependences of hyperfine parameters and their values allow us to attribute them to feroxyhyte—$\delta$-FeOOH [33–35].

**Table 2.** Results of non-model description of the Mössbauer spectra of samples.

| Temperature, K | | 296 | | | | | 78 | | | | |
|---|---|---|---|---|---|---|---|---|---|---|---|
| Sample | # | $\delta$ ($\delta_{ext}$) [1] | $\Delta$ ($\Delta_{ext}$) mm/s | $\Gamma_{exp}$ | $H_{ext}$ kOe | S# % | $\delta$ ($\delta_{ext}$) | $\Delta$ mm/s | $\Gamma_{exp}$ | $H_{ext}$ kOe | S# % |
| $Fe_3O_4$ | 1 | 0.317 | −0.02 | 0.42 | 464 | 64.8 | 0.445 | 0.08 | 0.45 | 514 | 66.7 |
| | 2 | 0.351 | 0.640 | 0.55 | - | 35.2 | 0.437 | −0.35 | 0.50 | 498 | 33.3 |
| $\gamma$-$Fe_2O_3$ | 1 | 0.330 | 0.00 | 0.43 | 461 | 67.7 | 0.447 | 0.069 | 0.48 | 513 | 66.7 |
| | 2 | 0.353 | 0.617 | 0.56 | - | 32.3 | 0.444 | −0.31 | 0.48 | 493 | 33.3 |
| $\delta$-FeOOH | 1 | (0.358) | (−0.162) | 0.48 | 437 | 94 | 0.489 | 0.02 | 0.46 | 512 | 50.0 |
| | 2 | 0.38 | 0.64 | 0.75 | - | 6 | 0.479 | −0.24 | 0.42 | 510 | 50.0 |
| $5Fe_2O_3 \cdot 9H_2O$ | 1 | 0.351 | 0.90 | 0.57 | - | 72 | (0.32) (0.38) | −0.13 | 1.08 | 387 114 | 49 |
| | 2 | 0.361 | 0.55 | 0.34 | - | 28 | 0.457 | 0.879 | 1.08 | - | 51 |
| $\alpha$-FeOOH-$Fe^{2+}$ | 1 | 0.391 | −0.233 | 0.343 | 335 | 80.8 | 0.473 | −0.25 | 0.295 | 483 | 82.4 |
| | 2 | 0.361 | 0.547 | 0.343 | - | 12.6 | 0.49 | 0.83 | 0.56 | - | 7.1 |
| | 3 | 1.16 | 2.28 | 0.82 | - | 6.6 | 1.42 | 2.81 | 0.82 | - | 10.5 |

[1] $\delta$ is the isomeric shift, $\Delta$ is the quadrupole splitting, $\Gamma_{exp}$ is the linewidth, $H_{ext}$ is the hyperfine magnetic field for the extremum of the mode distribution functions, and S# is the relative area of the subspectrum#.

At a temperature of 295 K, the Mössbauer spectrum of the $5Fe_2O_3 \cdot 9H_2O$ sample is paramagnetic, in accordance with Figure 2, and can be satisfactorily described by a pair of doublets corresponding to iron (+3) atoms in the octahedral oxygen environment indicated in Table 2 [36]. As the temperature decreases to 78 K, the width of the doublet increases significantly with the formation of extended "wings", which can be conditionally described by the broadened doublet and the distribution of the probability function of the isomer shift, and the magnetic field strength for the sextet, as shown in Figure 2. The calculated hyperfine MS parameters can correspond to akageneite—$\beta$-FeOOH [33,37] and ordered ferrihydrite [38,39].

The MS of the $\alpha$-FeOOH-$Fe^{2+}$ sample at different temperatures can be satisfactorily described by a superposition of two doublets and a sextet described by the probability function of the distribution of magnetic fields shown in Figure 2. The doublet with a larger isomer shift and quadrupole splitting (#3, Table 2) obviously corresponds to hydrated iron ions (+2) [36,40], and the doublet with a smaller isomeric shift and quadrupole splitting (#2, Table 2) to iron (+3) ions in an octahedral oxygen environment [36]. The temperature dependence of the hyperfine parameters for the ferromagnetic part of the spectrum (#1, Table 2) allows for unambiguously attributing it to goethite—$\alpha$-FeOOH with $Fe^{2+}$ [31,35,41]. Furthermore, the cipher $\alpha$-FeOOH-$Fe^{2+}$ for $\alpha$-FeOOH.

### 3.2. Magnetic Characteristics of Samples

The magnetization curves of the samples are shown in Figure 3. The absence of magnetic hysteresis in the $Fe_3O_4$ and $\gamma$-$Fe_2O_3$ samples indicates the production of superparamagnetic particles. It can be seen from the hysteresis curves that the $\delta$-FeOOH and $5Fe_2O_3 \cdot 9H_2O$ samples demonstrate linear behavior in the measured range of the magnetic field and exhibit paramagnetic properties. The $\alpha$-FeOOH-$Fe^{2+}$ sample exhibits non-linear

behavior near H = 0. Such behavior may indicate an additional magnetic phase or indicate the presence of unbalanced spins in the sample [42].

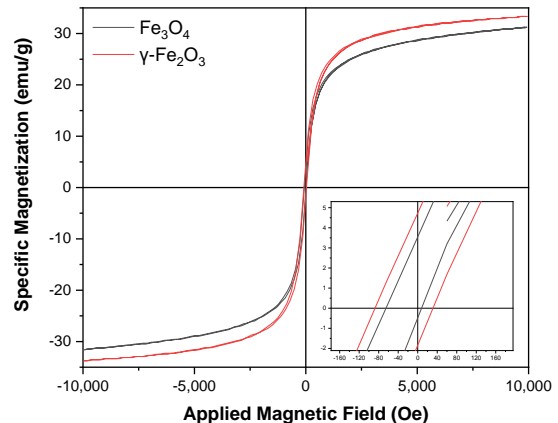
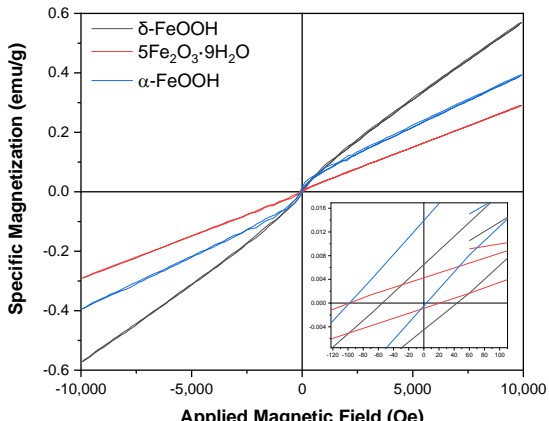

**Figure 3.** Magnetization curves of iron NP.

The results of the study of the magnetic characteristics of the samples in accordance with Table 3 indicate that the samples $Fe_3O_4$ and $\gamma$-$Fe_2O_3$ have the highest value of saturation magnetization. All samples have low magnetic characteristics compared to the data [43–47].

**Table 3.** Magnetic parameters.

| Sample | Magnetization Saturation ($M_s$) emu/g | Residual Magnetization ($M_r$) emu/g | Coercive Power ($H_c$) Oe |
|---|---|---|---|
| $Fe_3O_4$ | 31.5 | 2.0 | 39.5 |
| $\gamma$-$Fe_2O_3$ | 33.5 | 3.28 | 59.0 |
| $\delta$-FeOOH | 0.54 | 0.006 | 50.2 |
| $5Fe_2O_3 \cdot 9H_2O$ | 0.28 | 0.002 | 64.8 |
| $\alpha$-FeOOH-$Fe^{2+}$ | 0.41 | 0.007 | 50.1 |

The high values of the coercive force of the samples $\delta$-FeOOH, $5Fe_2O_3 \cdot 9H_2O$ and $\alpha$-FeOOH compared to $Fe_3O_4$ and $\gamma$-$Fe_2O_3$ can be associated with an asymmetric crystal lattice, as shown by XRD, which also increases the magnetic exchange anisotropy. According to Mössbauer spectroscopy data, the $5Fe_2O_3 \cdot 9H_2O$ sample can contain the $\beta$-FeOOH phase, and the $\alpha$-FeOOH-$Fe^{2+}$ sample contains hydrated iron ions (+2) and iron ions (+3) in an octahedral oxygen environment, which also increases the magnetic exchange anisotropy.

### 3.3. Textural Characteristics

Adsorption/desorption isotherms of nitrogen for iron nanoparticles are presented in Figure 4. All samples are characterized by type IV isotherms according to the classification, which indicates the occurrence of polymolecular adsorption and the presence of capillary condensation in mesopores. Samples of $Fe_3O_4$, $\gamma$-$Fe_2O_3$, and $\delta$-FeOOH have an H1-type hysteresis loop associated with the filling of mesopores due to capillary condensation. The $\alpha$-FeOOH-$Fe^{2+}$ sample has a hysteresis loop of the H4 type. The $5Fe_2O_3 \cdot 9H_2O$ has a hysteresis loop of the H2(a) type with a corpuscular structure, but the distribution and shape of the pores are inhomogeneous in this case. The presence of a steep rise in the isotherm at low pressures may indicate the presence of micropores.

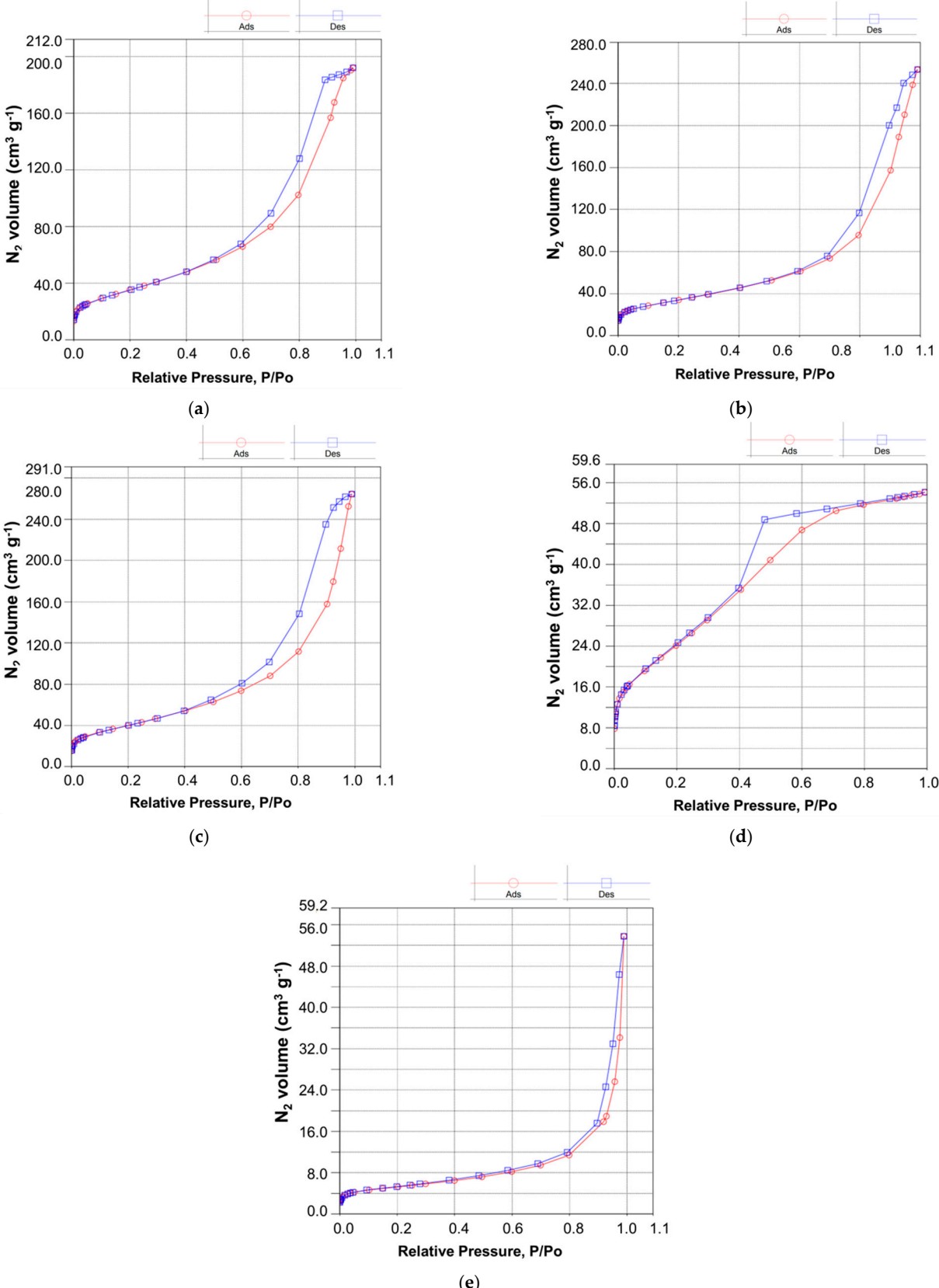

**Figure 4.** Isotherms of low-temperature (77 K) nitrogen adsorption/desorption of samples: (**a**) $Fe_3O_4$, (**b**) $\gamma$-$Fe_2O_3$, (**c**) $\delta$-FeOOH, (**d**) $5Fe_2O_3 \cdot 9H_2O$, (**e**) $\alpha$-FeOOH-$Fe^{2+}$.

Data for the specific surface area of the samples by the BET method and the average pore diameter by the BJH method (Table 4 and see Supplementary Materials, Figure S1) demonstrated that all NPs have a significant decrease in the specific surface area (SSA), compared with the data [25,48–50]. Samples of feroxyhyte and ferrihydrite showed a significant decrease in specific surface area compared to the data [25,47] as a result of the synthesis conditions we used.

**Table 4.** Textural parameters.

| Sample | BET [1] | | BJH [2] | |
|---|---|---|---|---|
| | SSA m$^2$/g | Pore Volume cm$^3$/g | Pore Volume cm$^3$/g | Pore Diameter nm |
| $Fe_3O_4$ | 130.9 | 0.30 | 0.29 | 8.86 |
| $\gamma$-$Fe_2O_3$ | 123.0 | 0.39 | 0.39 | 8.71 |
| $\delta$-FeOOH | 147.5 | 0.41 | 0.41 | 8.94 |
| $5Fe_2O_3 \cdot 9H_2O$ | 97.0 | 0.084 | 0.056 | 3.54 |
| $\alpha$-FeOOH-$Fe^{2+}$ | 18.8 | 0.083 | 0.080 | 23.8 |

[1] Brunauer–Emmett–Teller, [2] Barrett–Joyner–Halenda.

### 3.4. Iron Ions Release

Various iron-based nanomaterials have been widely investigated to induce ferroptosis for the proposed reason that the Fenton reaction between iron ions and tumor cellular $H_2O_2$ could generate reactive oxygen species (ROS), which further causes the accumulation of lipid peroxidation [8,13,51–55]. Iron-containing compounds such as magnetite [12,56], maghemite [5], and iron salts [51,52] were used as sources of $Fe^{2+}$ and $Fe^{3+}$ as a catalyst for the Fenton reaction. A disadvantage of the latter is the impossibility of the application of a magnetic field for the target drug delivery.

UV-Vis spectroscopy was used to demonstrate the kinetics and concentration of the released $Fe^{2+}$ and $Fe^{3+}$ from nanoparticles $Fe_3O_4$, $\gamma$-$Fe_2O_3$, $\delta$-FeOOH, $5Fe_2O_3 \cdot 9H_2O$, $\alpha$-FeOOH-$Fe^{2+}$ (see Supplementary Materials, Figure S2). The concentrations of iron ions were recalculated per 1 g of the iron.

According to complexometric results (Figure 5), $\alpha$-FeOOH-$Fe^{2+}$ released both $Fe^{2+}$ and $Fe^{3+}$, while $\delta$-FeOOH, $Fe_3O_4$, $\gamma$-$Fe_2O_3$, and $5Fe_2O_3 \cdot 9H_2O$ released only $Fe^{3+}$. In this case, the concentration of ions released by nanoparticles differs. Thus, the concentration of ions released by all nanoparticles, except $\alpha$-FeOOH-$Fe^{2+}$, reached 9.1 mg/L for $Fe^{3+}$, which makes them preferable for controlling the catalysis of the Fenton reaction. While the high concentration of iron ions to 90 mg/L for $Fe^{3+}$ and 316 mg/L for $Fe^{2+}$ released from $\alpha$-FeOOH-$Fe^{2+}$ can be used in case of aid therapy. Moreover, the kinetics of the release of the total concentration of ions is similar to all iron-containing compounds: the concentration of ions in the solution increases within 3 h. After 24 h, a decrease in the concentration of ions is observed, which may indicate the adsorption of ions from the solution onto the NPs' surface [57].

The estimated release from composites produced by the lowest amount of $Fe^{3+}$ was released from glass compositions with the highest $Fe_2O_3$ content [5]. It can be suggested that the release of $Fe^{3+}$ was dictated by the durability of the glass rather than the amount of $Fe^{3+}$ available in the glass. The character of the release of ions is similar to the one from the nanoparticles we studied: the authors observed an increase in the concentration of the released nanoparticles within 24 h and then a significant, down to 0 mg/L, decrease in the concentration within 95 h for nanoparticles with different concentrations of $Fe_2O_3$. In our case, the low concentration of released ions at pH close to neutral (6.6 for $\gamma$-$Fe_2O_3$ and 7.0 for $Fe_3O_4$) can be associated with the formation of $Fe^{3+}$ hydroxocomplexes under these conditions. At the same time, the low concentration of released ions makes the usage of magnetic nanoparticles more controllable, for example, by adding $H_2O_2$. The high release

of ions from $\alpha$-FeOOH-$Fe^{2+}$ can lead to a speedy Fenton reaction that can be used for aid therapy cases. A comparison of the concentration of $Fe^{3+}$ released by $Fe_3O_4$ and $\gamma$-$Fe_2O_3$ showed that the nanoparticles release approximately the same concentration of ferric ions into the solution, which confirms the identity of their structural parameters shown by XRD and surrounding local data for Fe atoms/ions shown by Mössbauer spectroscopy. The concentration of $Fe^{2+}$ released by $Fe_3O_4$ may be too low to be detected by the method used, or $Fe^{2+}$ at a pH close to neutral (6.6 for $\gamma$-$Fe_2O_3$ and 7.0 for $Fe_3O_4$) is rapidly oxidized to $Fe^{3+}$.

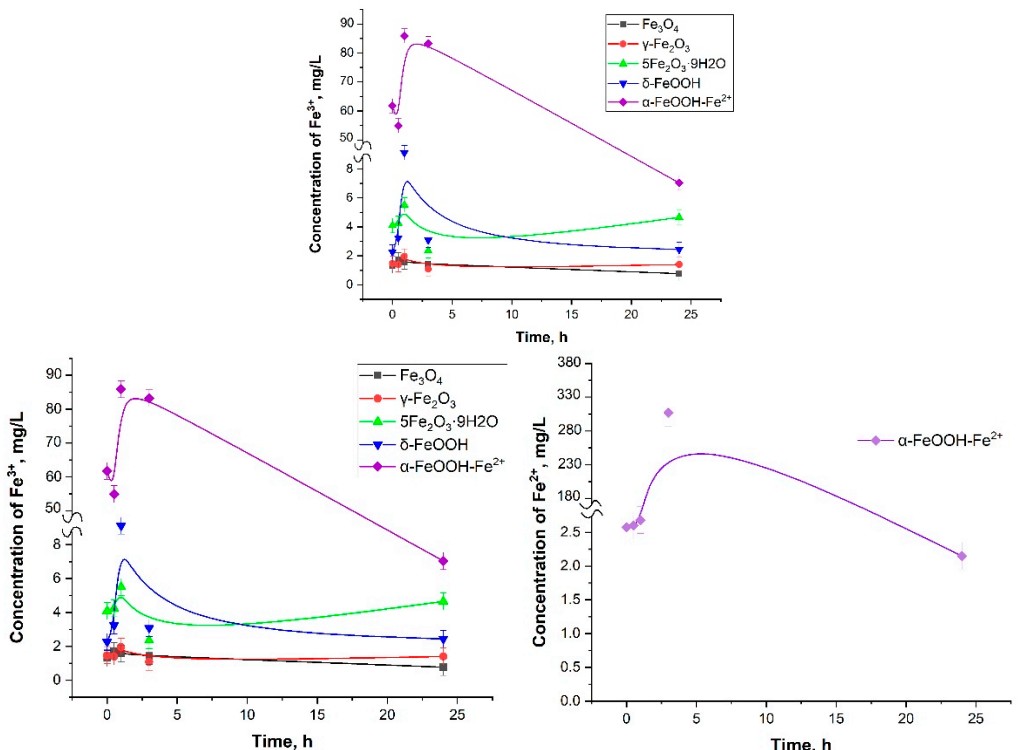

**Figure 5.** $Fe^{2+}$ and $Fe^{3+}$ ion release kinetics from $Fe_3O_4$, $\gamma$-$Fe_2O_3$, $\delta$-FeOOH, $5Fe_2O_3 \cdot 9H_2O$, $\alpha$-FeOOH nanoparticles ($Fe_3O_4$ pH 7, $\gamma$-$Fe_2O_3$ pH 6.6 $\alpha$-FeOOH-$Fe^{2+}$ pH 6.9 $\delta$-FeOOH pH 6.7, $5Fe_2O_3 \cdot 9H_2O$ pH 6.9) (see Supplementary Materials Figure S2).

Comparative assessment of the concentration of ions released by various iron-containing materials is complicated by several factors: (1) different methods of assessment (ICP-MS or UV-Vis spectroscopy with different reagents), which imply different errors, and (2) no information about the processing of primary data in terms of iron content in the sample, the concentration of trace elements, etc.

Evaluation of ions released for $\delta$-FeOOH revealed that the concentration of ferrous irons increases strictly after three hours of suspension: from 2.2 mg/L to 9.1 mg/L for $Fe^{3+}$. Further, the concentration of ions in the solution decreases to 2.4 mg/L for $Fe^{3+}$.

The study of $5Fe_2O_3 \cdot 9H_2O$ as a source of iron ions showed that the concentration of $Fe^{3+}$ in the solution remains practically unchanged during the first hour of the measurements (4–5.4 mg/L), then decreases to 2.3 mg/L and increases again to 4.6 mg/L after 24 h. These processes may indicate ongoing adsorption and desorption of ions from the surface of nanoparticles.

Withal, $\alpha$-FeOOH-$Fe^{2+}$ is released at least 50 times more $Fe^{3+}$ and 150 times more $Fe^{2+}$ than the rest of the studied compounds, although its surface area was 18.2 $m^2$/g, which is much less than the surface area of the other compounds. The high concentration of $Fe^{2+}$ in the solution is explained by the modified synthesis of $Fe^{2+}$-doped $\alpha$-FeOOH in ambient conditions resulting in the $Fe^{2+}$ presence. The concentration of $Fe^{3+}$ increased from ~60 mg/L to ~80 mg/L after an hour and then decreased to 35.1 mg/L after a day

of observation; the concentration of $Fe^{2+}$ increased sharply after 3 h from ~180 mg/L to 316 mg/L and then also decreased to 150 mg/L. In general, goethite released ~3 times more ferrous ions than ferric iron during the day.

## 4. Conclusions

The estimation of the most prospective iron oxide nanoparticles in terms of release of $Fe^{2+}/Fe^{3+}$, surface, and magnetic characteristics was performed in order to further utilize them as mediators for the Fenton reaction. The crystal structure and oxygen local environment of all samples that were analyzed using XRD and Mossbauer spectroscopy, respectively, correspond to the mixture of $Fe_3O_4$ and $\gamma$-$Fe_2O_3$, $\gamma$-$Fe_2O_3$, $\delta$-FeOOH, $5Fe_2O_3 \cdot 9H_2O$, and $Fe^{2+}$-doped $\alpha$-FeOOH with $Fe^{2+}$. Nanoparticles of $Fe_3O_4$ and $\gamma$-$Fe_2O_3$ synthesized in ambient conditions are a mixture of magnetite and maghemite with composition as $Fe_{2\cdot66}O_4$, and goethite with $Fe^{2+}$, respectively. The saturation magnetization for $Fe_3O_4$ (31.6 emu/g) and $\gamma$-$Fe_2O_3$ (33.8 emu/g), as well as the surface area of these particles (130 and 123 $m^2$/g), make it possible to consider them as promising magnetically controlled carriers. A low release of $Fe^{2+}/Fe^{3+}$ from nanoparticles during the first three hours makes them preferable for controlling the catalysis of the Fenton reaction. Whereas a high concentration of iron ions to 90 mg/L for $Fe^{3+}$ and 316 mg/L for $Fe^{2+}$ released from complex $\alpha$-FeOOH-$Fe^{2+}$ allows it to be utilized as an aid therapy agent. Thus, primary results on $Fe^{2+}/Fe^{3+}$ release by iron oxide and hydroxide nanoparticles will provide information to form a whole new class of prospective preparations that will express maximal efficacy and specificity in ferroptosis-inducing systems.

**Supplementary Materials:** The following supporting information can be downloaded at: https://www.mdpi.com/article/10.3390/magnetochemistry9010003/s1, Figure S1: Pore size distribution from a BJH model of samples: (a) $Fe_3O_4$, (b) $\gamma$-$Fe_2O_3$, (c) $\delta$-FeOOH, (d) $5Fe_2O_3 \cdot 9H_2O$, (e) $\alpha$-FeOOH; Figure S2: UV-Vis spectra of $Fe^{2+}$ and $Fe^{3+}$ ion release kinetics (a) $Fe_3O_4$, (b) $\gamma$-$Fe_2O_3$, (c) $\delta$-FeOOH, (d) $5Fe_2O_3 \cdot 9H_2O$, (e) $\alpha$-FeOOH nanoparticles.

**Author Contributions:** Conceptualization, K.K. and L.B.; methodology, L.B. and K.K.; software, L.B. and L.T.; validation, N.T. and G.D.; formal analysis, D.P.; investigation, A.D., M.P., L.B. and D.P.; resources, G.D. and N.T.; data curation, S.J.; writing—original draft preparation, writing—review and editing, K.K. and L.B.; supervision, S.J. All authors have read and agreed to the published version of the manuscript.

**Funding:** This research was funded by Russian Science Foundation, N 22-73-10222.

**Institutional Review Board Statement:** Not applicable.

**Informed Consent Statement:** Not applicable.

**Data Availability Statement:** Not applicable.

**Conflicts of Interest:** The authors declare no conflict of interest.

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
