# Peer review of "Iron Oxides Nanoparticles as Components of Ferroptosis-Inducing Systems: Screening of Potential Candidates"

_magnetochemistry, doi:10.3390/magnetochemistry9010003_

Round 1
Reviewer 1 Report
Review
Line 24: Why is it decreasing? Does it recrystallize into more stable forms (alfa-Fe2O3?) or just to larger crystals?
Line 28: Why does stable alpha oxyhydroxide dissolve more easy than others? And formally it is all iron in Fe(+3) state. So why the concentration of Fe(+2) is so high and is much larger than Fe(+3)?
Line 28: "316 mg/L for Fe2+ " but it is only 236 mg/L according to Fig. 5. Where are correct data?
Line 40, 43: The first explanation of abbreviation "reactive oxygen species (ROS)" should be introduced at first its appearance (line 40 instead of line 43), not at the third appearance (line 79).
Line 57: remove "in" in "prepared in in vitro".
Line 86: "α-FeOOH - due to low oxidation in acid medium" does it mean oxidation to Fe(+4)?! Please give more details.
Line 97: "All preparations were chemically pure." Probably it means "All reagents were chemically pure" or "All precursors were chemically pure"?
Compare with the next line 98: "2.2. Preparation of Fe3O4 ".
Line 103: "100 mL of 10% NaOH (pH = 10) was added." Please calculate pH correct.
Line 104: "magnet (Nd; 0.3 Tc)" probably means "magnet (Nd; 0.3 T)".
Line 105: "To remove Na+, Cl–, the precipitate was washed" better replace with "To remove Na+, Cl–, SO4-2, OH-, the precipitate was washed ".
Line 115: "stirred on amechanical stirrer" need to change to "stirred on a mechanical stirrer".
Line 115: to change "A green solution" to "A green suspension" or "A green colloidal solution".
Line 128: the α-FeOOH is a Fe(+3) compound. The described procedure likely to produce Fe(OH)2.
Please indicate the color of alfa-FeOOH you obtain.
For instance in [https://doi.org/10.1007/s11051-008-9467-z] authors uses Fe(+3) nitrate:
"Precipitation of α-FeOOH in aqueous solution was performed by placing 0.025 mol dm−3 solution of [Fe(NO3)3 · 4H2O—BDH] at pH 2 overnight in a water-bath set at 50 °C. The pH of the solution was adjusted to 2 using nitric acid. A yellow precipitate formed and separated from the acidic solution by centrifugation."
Please put the references to the method you used in opposition to mentioned above.
Line 138: "Diffraction patterns and analysis XRF data were obtained using the Match! and OriginPro software. " better to use "Powder X-Ray analysis was done using Match! software".
Line 149: "the samples were degassed “thermal training”" better to use "the samples were degassed".
Line 150: "heating in a stationary nitrogen flow in a vacuum". So is it vacuum or nitrogen flow? If there is nitrogen flow, then please put the approximate nitrogen pressure: 1 bar, 1mm Hg or 1 Pa?
Line 150: "a temperature of 150 ° C" in vacuum will turn FeOOH to Fe2O3.
Lines 158-164 very poor experiment description: line 159: "in buffer solutions" - please indicate components and concentrations. Line 159-160: "in buffer solutions containing potassium thiocyanate (50% solution) and HCl (18.25% solution)... and potassium hexacyanoferrate(III) (1% solution)". What is the aim of buffer solution if next to that you adding 18% HCl? Please put the reason for HCl addition and volumes of KSCN and HCl solutions. Please indicate the target pH upon adding HCl.
Importance of that you underline on lines 298-300 etc.
"At different time points (0, 0.5, 1, 3 and 24 h), the mixtures were centrifuged" - does it mean, that oxyhydrate nanoparticles, buffer, KSCN, HCl, K3[Fe(CN)6] all were in single flask? Is it was stirred?
Lines 167-170: the ferrihydrite is claimed as weakly crystalline. And the other phases are well-crystallized. It is correct. Contrary to that is content of Table 1. In the Table 1 the Fe3O4 and gamma-Fe2O3 are both have smaller crystallite size.
Fig. 1: the (c) and (d) patterns contain strong extra reflections.
Line 176-177: "Table 1 presents the lattice parameters of each sample and the for each refinement: Good-176 ness of Fit (GOF), which is the RWP/Rexp ratio [27]." better to be "Table 1 presents the lattice parameters of each sample and the Goodness of Fit (GOF), which is the RWP/Rexp ratio [27]."
There is no GOF parameter in the Table 1!
Table 1: what does it mean "Structure Fe2.66O4 Fe2.56O4"? How the numbers 2.66 and 2.56 were obtained? Please put in the manuscript.
The Fe2.56O4 must have Fe(+4).
Why magnetite and gamma-Fe2O3 have certain composition like Fe2.66O4 and the other nanomaterials are not? Please put in all.
Why Fe3O4 composition is so far from Fe3O4 and is Fe2.66O4?
What is the method for "Composition determined"? Please put in the manuscript.
Why uncertainty of crystallite size have only Fe3O4 and gamma-Fe2O3, but the others have not?
Lines 184-198: the Fe3O4 in fact is gamma-Fe2O3.
Further discussion of titration results should keep it in focus.
Line 224: "akaganite" to replace with "akaganeite".
Table 3: please check the coercive power for ferrihydrite and goetite.
Line 259-560: "the presence of a steep rise in the isotherm at low pressures may indicate the presence of micropores".
That feature mentioned for alfa-FeOOH, but it is the smallest one among the others.
Please reformulate to underline larger micropore area in the other samples.
Line 286: the Supplementary materials doesn't contain Table S2. Please put it or change to Fig. S2.
Line 287: "As the results of complexometric reactions showed (Figure 5)".
The Fig. 5 do not contain reactions. Better to use "According to complexometric results (Fig. 5)".
Line 292: "316 mg/L for Fe2+ " but it is only 236 mg/L according to Fig. 5. Where are correct data?
Line 293, 373: "can lead to uncontrolled catalysis way". Please explain what do you mean? Larger catalyst concentration in a cancer cell may produce more than 100% of ROS from same amount of H2O2? Isn't a hydrogen peroxide indeed a limitation factor for more ROS production in a cell?
Moreover, looking on fast decay of Fe concentration isn't it better to use longlasting effect of alfa-FeOOH ruther than the others?
Lines 295-297: "the concentration of ions in the solution increases for 3 hours. After 24 hours a decrease in the concentration of ions is observed, which may indicate the adsorption of ions from the solution onto the NPs surface".
So what was originally the driving force of Fe ions desorption for the first 3 hours? Following your idea, after 3 hours, Fe ions realize that they take a wrong decision to desorb from surface?
Lines 298-318: since there are shell formation in composite (or micelle), these references are hard to compare to discussed results. What shell on maggemite NP do you expect in your compexation experiment?
Lines 319-320: "practically insoluble Fe(OH)3 complexes from Fe3+ ions under these conditions." But the Fe3+ ions you mention in that sentence come from what source? From much more soluble NP? The "Fe(OH)3 complexes" are molecular uncharged species? Than why it is insoluble? The Si(OH)4 molecule for instance, is soluble. Such a species are highly reactive and may react with ligands like SCN-.
Please give the references to approve your point.
Line 322: reformulate "release of α-FeOOH ions".
Line 326: "their microstructure, shown by Mössbauer spectroscopy and XRD". The therm "microstructure" looks unsuitable. XRD provide structural parameters, Mössbauer spectroscopy provide local surrounding data for Fe atoms/ions.
Please select proper therm.
Line 328: "at a pH close to neutral (6.6 for γ-Fe2O3 and 7.0 for Fe3O4)".
Line 350: "the pH of the solution, which was 3.45".
So what was the pH of the solutions you used for testing of NP?
Please put description of compleximetric experiment (see in the beginning lines 158-160):
please indicate components and concentrations. Line 159-160: "in buffer solutions containing potassium thiocyanate (50% solution) and HCl (18.25% solution)... and potassium hexacyanoferrate(III) (1% solution)". What is the aim of buffer solution if next to that you adding 18% HCl? Please put the reason for HCl addition and volumes of KSCN and HCl solutions. Please indicate the target pH upon adding HCl.
Importance of that you underline on lines 298-300 etc.
"At different time points (0, 0.5, 1, 3 and 24 h), the mixtures were centrifuged" - does it mean, that oxyhydrate nanoparticles, buffer, KSCN, HCl, K3[Fe(CN)6] all were in single flask?
Line 328: "Fe2+ ... are rapidly oxidized to Fe3+ ". Please explain why it is not oxidized in case of delta-FeOOH? Isn't Fe2+ more rapidly reacted with 1% solution K3[Fe(CN)6] rather than with O2?
Say, (NH4)2Fe(SO4)2 solution is stable enough for hours in air and was widely used for titrimetric analysis for 1800-1950 period.
See also line 340: "Fe3+/Fe2+ remains almost unchanged and close to 5 during the entire observation time", so the concentrations changes 3-4 times, but the oxidation of Fe(+2) is not detected.
See also line 349.
Line 333-334: poor English.
Line 344: the therms "adsorption and desorption of ions" is much more preferable instead of "ion release" (see line 336 and others).
"Release" is well applicable to the well soluble substance, like "release of ascorbic acid from gelly capsula" or "release of HCN from apricot shell".
"Release" is non-reversible process.
The nonstationary process you discuss is reversible and may be connected to adsorption and desorption.
Line 346: "50 times more Fe3+ and 150 times less Fe2+ than" to replace with "50 times more Fe3+ and 150 times more Fe2+ than".
Line 354: "316 mg/L" but it is only 236 mg/L according to Fig. 5. Where are correct data?
Conclusions: poor English. Please reformulate.
Line 362: "mixture of magnetite and maghemite with composition as Fe2.66O4 instead of Fe3O4."
I doesn't see evidence for mixture. According to Moessbauer it is maghemite. It also maghemite according to Fe(+2) complexometric study. PXRD data and magnetic measurements are fit well with maghemite.
Do you have EXAFS or XANES data to approve Fe(II) in that sample?
Line 372: "316 mg/L" but it is only 236 mg/L according to Fig. 5. Where are correct data?
Fig. S2 is hard to read. Numbers are small, lines are of same color, lines are too close to each other. Please remove part of lines, change the line type (dot, dash-dot) and its colors and thickness to make graph easy to read.
The absorbance at Fig. S2(e) is very similar or smaller to (c) and (d). From manuscript Fig. 5 the α-FeOOH (e) is orders of magnitude larger values of Fe ions release comparing to (c) and (d). Please explain it and make it clear in the text. For instance, put mass of the samples used on Fig. 2S.
For the Fig. S2 it is recommended to present additional graph (f) with Fe(+3)-SCN complex spectrum and separately Prussian blue spectrum.

Author Response
Dear Reviewer,
Thank you for your detailed review and constructive comments.
Please find below our response to your notes.
|
1. |
Line 24: Why is it decreasing? Does it recrystallize into more stable forms (alfa-Fe2O3?) or just to larger crystals? |
Dr. Prof. Scherer in various publications describes the sorption-desorption of iron ions in relation to iron oxides and oxyhydroxides. In particular, they described the mechanisms of Fe(II) sorption on the surface of Goethite and atom exchange between goethite and aqueous Fe(II) [Handler, R. M., Beard, B. L., Johnson, C. M., & Scherer, M. M. (2009). Atom Exchange between Aqueous Fe(II) and Goethite: An Fe Isotope Tracer Study. Environmental Science & Technology, 43(4), 1102–1107. doi:10.1021/es802402m]. The authors showed complex mechanism of interaction between ions in solution and iron oxides and oxyhydroxides showing that “sorption of Fe(II) onto goethite results in electron transfer between the sorbed Fe(II) and the structural Fe(III) in goethite”. The approach used in this study is similar to that followed by Skulan et al. (27), Poulson et al. (28), Welch et al. (29), and Shahar et al. (30) to identify isotope exchange between aqueous Fe(III) and hematite, aqueous Fe(III) and ferrihydrite, aqueous Fe(III) and Fe(II), and magnetite and fayalite, respectively. Although sorption of Fe(II) to goethite and hematite occurs over a wide range of pH values (5, 16), and interfacial electron transfer has been demonstrated on these oxides (10, 17), no phase transformations have been observed, and little to no reductive dissolution of Fe(II) occurs after reaction of Fe(II) with goethite or hematite, respectively (12). Pedersen et al. incorporated 55Fe into several different Fe oxides and observed the release of 55Fe into solution upon exposure to aqueous Fe(II) (12). Release of 55Fe into solution was observed for lepidocrocite, ferrihydrite, and goethite, but not for hematite. On the other side, the observed decrease in the concentration of ions within 24 hours may be due not to the sorption of ions, but to the formation of insoluble hydroxocomplexes Fe(OH)3 and Fe(OH)2+, which are not detected by the selected complexometric methods. |
|||||||||||||||||||||||||||||||||||||||||||||||||||||||||||||||||||||
|
2. |
Line 28: Why does stable alpha oxyhydroxide dissolve more easy than others? And formally it is all iron in Fe(+3) state. So why the concentration of Fe(+2) is so high and is much larger than Fe(+3)?
|
In fact, we propose to synthesize the ZVI nanoparticles and use appropriate protocol for the nanoZVI. But, the difference was in ambient conditions due to economical reason in order to scale up this synthesis. According to the XRD analysis the goethite phase was identified, the Mössbauer spectra detected goethite and the presence of Fe2+ ions. Therefore, air-environment synthesis of nanoZVI led to the formation of mixture of alpha-FeOOH and Fe2+ and high release of Fe2+.
|
|||||||||||||||||||||||||||||||||||||||||||||||||||||||||||||||||||||
|
3. |
Line 28: "316 mg/L for Fe2+ " but it is only 236 mg/L according to Fig. 5. Where are correct data?
|
316 mg/L is the correct data. We have changed the size of the axis so that the point gets on the figure. |
|||||||||||||||||||||||||||||||||||||||||||||||||||||||||||||||||||||
|
4. |
Line 40, 43: The first explanation of abbreviation "reactive oxygen species (ROS)" should be introduced at first its appearance (line 40 instead of line 43), not at the third appearance (line 79).
|
Thanks, you are right, done. |
|||||||||||||||||||||||||||||||||||||||||||||||||||||||||||||||||||||
|
5. |
Line 57: remove "in" in "prepared in in vitro".
|
Thanks, done |
|||||||||||||||||||||||||||||||||||||||||||||||||||||||||||||||||||||
|
6. |
Line 86: "α-FeOOH - due to low oxidation in acid medium" does it mean oxidation to Fe(+4)?! Please give more details.
|
||||||||||||||||||||||||||||||||||||||||||||||||||||||||||||||||||||||
|
7. |
Line 97: "All preparations were chemically pure." Probably it means "All reagents were chemically pure" or "All precursors were chemically pure"?
|
Yes, we have corrected |
|||||||||||||||||||||||||||||||||||||||||||||||||||||||||||||||||||||
|
8. |
Compare with the next line 98: "2.2. Preparation of Fe3O4 ".
|
Ok, done |
|||||||||||||||||||||||||||||||||||||||||||||||||||||||||||||||||||||
|
9. |
Line 103: "100 mL of 10% NaOH (pH = 10) was added." Please calculate pH correct.
|
While stirring on a mechanical stirrer (1000 rpm) 10% NaOH was added to pH 10 |
|||||||||||||||||||||||||||||||||||||||||||||||||||||||||||||||||||||
|
10. |
Line 104: "magnet (Nd; 0.3 Tc)" probably means "magnet (Nd; 0.3 T)".
|
Yes, we have corrected |
|||||||||||||||||||||||||||||||||||||||||||||||||||||||||||||||||||||
|
11. |
Line 105: "To remove Na+, Cl–, the precipitate was washed" better replace with "To remove Na+, Cl–, SO4-2, OH-, the precipitate was washed ".
|
Yes, we have corrected |
|||||||||||||||||||||||||||||||||||||||||||||||||||||||||||||||||||||
|
12. |
Line 115: "stirred on a mechanical stirrer" need to change to "stirred on a mechanical stirrer".
|
Thanks, done |
|||||||||||||||||||||||||||||||||||||||||||||||||||||||||||||||||||||
|
13. |
Line 115: to change "A green solution" to "A green suspension" or "A green colloidal solution".
|
Thanks, done |
|||||||||||||||||||||||||||||||||||||||||||||||||||||||||||||||||||||
|
14. |
Line 128: the α-FeOOH is a Fe(+3) compound. The described procedure likely to produce Fe(OH)2. |
We used the modified method for synthesis of nanoZVI in ambient conditions (see item 17)
|
|||||||||||||||||||||||||||||||||||||||||||||||||||||||||||||||||||||
|
15. |
Please indicate the color of alfa-FeOOH you obtain. For instance in [https://doi.org/10.1007/s11051-008-9467-z] authors uses Fe(+3) nitrate: "Precipitation of α-FeOOH in aqueous solution was performed by placing 0.025 mol dm−3 solution of [Fe(NO3)3 · 4H2O —BDH] at pH 2 overnight in a water-bath set at 50 °C. The pH of the solution was adjusted to 2 using nitric acid. A yellow precipitate formed and separated from the acidic solution by centrifugation."
|
A yellow precipitate is formed and separated from the acidic solution by centrifugation. |
|||||||||||||||||||||||||||||||||||||||||||||||||||||||||||||||||||||
|
16. |
|||||||||||||||||||||||||||||||||||||||||||||||||||||||||||||||||||||||
|
17. |
Please put the references to the method you used in opposition to mentioned above.
|
Method for nano ZVI was used by modification of synthesis conditions, air instead of inert medium.
Lowry, G. V., & Johnson, K. M. (2004). Congener-Specific Dechlorination of Dissolved PCBs by Microscale and Nanoscale Zerovalent Iron in a Water/Methanol Solution. Environmental Science & Technology, 38(19), 5208–5216. doi:10.1021/es049835q
Into 1 L of a 30% (volume) MeOH/deionized water solution was dissolved 20 g of FeSO4‚7H2O. While stirring, 10 mL of 5 N NaOH aqueous solution was added dropwise to the dissolved iron solution, yielding a pH of 6.1. Next, 50 mL of a 2.1 M NaBH4 aqueous solution was added at a rate ∼0.5 mL/sec. This is approximately 2.9 times the stoichiometric amount of NaBH4 required to reduce all the dissolved Fe2+ in solution to zerovalent iron. A fine black precipitate formed instantly upon addition of NaBH4 and remained in solution throughout the process. The mixture was stirred for 20 min before being centrifuged in plastic centrifuge tubes for 5 min at 3500 rpm to separate the solids. The supernatant was discarded and the remaining solids were rinsed with MeOH twice to remove any excess salt. The nanoscale ZVI particles were then dried in a 106°C oven for 4 h under nitrogen gas. Once dry, the particles remained in the oven for a minimum of 8 h to cool and allow oxygen to slowly bleed into the oven. This passivated the iron surface slightly and prevented the highly reactive iron particles from spontaneously igniting. The dried particles were finely ground and stored in sealed serum bottles until use.
|
|||||||||||||||||||||||||||||||||||||||||||||||||||||||||||||||||||||
|
18. |
Line 138: "Diffraction patterns and analysis XRF data were obtained using the Match! and OriginPro software. " better to use "Powder X-Ray analysis was done using Match! software".
|
Thanks, done |
|||||||||||||||||||||||||||||||||||||||||||||||||||||||||||||||||||||
|
19. |
Line 149: "the samples were degassed “thermal training”" better to use "the samples were degassed".
|
Thanks, done |
|||||||||||||||||||||||||||||||||||||||||||||||||||||||||||||||||||||
|
20. |
Line 150: "heating in a stationary nitrogen flow in a vacuum". So is it vacuum or nitrogen flow? If there is nitrogen flow, then please put the approximate nitrogen pressure: 1 bar, 1mm Hg or 1 Pa?
|
We have refined the research methodology. The study was carried out in a He(helium) current, 150°C, 2 h, flow rate 0.2 l/h (the speed was determined by counting the bubbles in the liquid - 10 bubbles/min) |
|||||||||||||||||||||||||||||||||||||||||||||||||||||||||||||||||||||
|
21. |
Line 150: "a temperature of 150 °C" in vacuum will turn FeOOH to Fe2O3.
|
At this temperature and duration of heating (2 h), there will be no phase transition of FeOOH to Fe2O3. There will be desorption of only physically adsorbed water.
According to [Ponomar, V. P. (2018). Thermomagnetic properties of the goethite transformation during high-temperature treatment. Minerals Engineering, 127, 143–152. doi:10.1016/j.mineng.2018.08.016], “When heated under oxidative conditions, synthetic goethite and natural brown goethite exhibited two significant changes in weight, which meant the removal of absorbed water and OH groups. The transformation of goethite into hematite occurred at about 270 °C for the synthetic sample and at 310 °C for the natural brown sample”.
Also Valezi et al [Valezi, D. F., Baú, J. P. T., Zaia, D. A. M., Costa, A. C. S., Urbano, A., Tupan, L. F. S., … Di Mauro, E. (2019). Enhanced Magnetic Component in Synthetic Goethite (α‐FeOOH) and its Relation with Morphological and Structural Characteristics. Physica Status Solidi (b), 1800578. doi:10.1002/pssb.201800578] showed, that in case of goethite, TGA experiment demonstrated “The first, between room temperature and about 130 C, is mainly related to the adsorbed water mass loss (%H2O).[12,13,23–26] The second one, between 130 and 500 C, presents the most crucial mass loss stage. This mass loss stage is caused by a loss of structural hydroxyl (%OH), which results in the transformation of the goethite to hematite.[12,13,23–26]” |
|||||||||||||||||||||||||||||||||||||||||||||||||||||||||||||||||||||
|
22. |
Lines 158-164 very poor experiment description: line 159: "in buffer solutions" - please indicate components and concentrations. Line 159-160: "in buffer solutions containing potassium thiocyanate (50% solution) and HCl (18.25% solution) and potassium hexacyanoferrate (III) (1% solution)". What is the aim of buffer solution if next to that you adding 18% HCl? Please put the reason for HCl addition and volumes of KSCN and HCl solutions. Please indicate the target pH upon adding HCl.
|
We have added a more detailed description of the methodology:
To detect iron ions, powder were suspended in deinized water, then subjected to centrifugation (6000 rpm, 5 min) after 0, 0.5, 1, 3, and 24 hours. Appropriate ion detection reagents were added to the supernatant after separation. To detect Fe3+ ions, 200 μl of potassium thiocyanate KSCN (50% solution) and 200 μl of HCl (18.25% solution) were added to 5 ml of supernatant, since the reaction between the Fe3+ ions and KSCN proceeds in a strongly acidic medium at pH close to 2. Then the solution was kept for 20 minutes to reach equilibrium and the absorption spectrum in the region of 490 nm was measured. To detect Fe2+ ions, 2 ml phenanthroline C12H8N2∙H2O (2.5% solution) and 600 μl of ammonium acetate buffer solution (250 ml of NH4OH and 900 ml of glacial acetic acid) were also added to the 5 ml of supernatant. Further, the solution was also kept for 20 minutes to reach equilibrium (no color change) and examined in the region of 690 nm. The absorbance was detected by UV–Vis-NIR spectrophotometry (Cary UV-Vis-NIR Spectrophotometer, Agilent Technologies) |
|||||||||||||||||||||||||||||||||||||||||||||||||||||||||||||||||||||
|
23. |
Importance of that you underline on lines 298-300 etc.
|
These lines were deleted |
|||||||||||||||||||||||||||||||||||||||||||||||||||||||||||||||||||||
|
24. |
"At different time points (0, 0.5, 1, 3 and 24 h), the mixtures were centrifuged" - does it mean, that oxyhydrate nanoparticles, buffer, KSCN, HCl, K3[Fe(CN)6] all were in single flask? Is it was stirred?
|
According to the procedure, sample suspensions were stored for 0, 0.5, 1, 3, and 24 hours; then, the supernatant was separated by centrifugation. Reagents for the determination of iron ions (method line 159-170) were already added to the supernant. Solutions with reagents were examined 20 minutes after preparation. |
|||||||||||||||||||||||||||||||||||||||||||||||||||||||||||||||||||||
|
25. |
Lines 167-170: the ferrihydrite is claimed as weakly crystalline. And the other phases are well-crystallized. It is correct. Contrary to that is content of Table 1. In the Table 1 the Fe3O4 and gamma-Fe2O3 are both have smaller crystallite size.
|
We have made a mistake with size calculation. Corrected size was calculated and presented in revised version of paper. |
|||||||||||||||||||||||||||||||||||||||||||||||||||||||||||||||||||||
|
26. |
Fig. 1: the (c) and (d) patterns contain strong extra reflections.
|
The Math program contains a database where is a few data on feroxyhyte and ferrihydrite unlike magnetite or goethite. That’s why it was not possible to choose a better model. We gathered the date massive containing diffraction patterns for feroxyhyte and ferrihydrite. It can be seen that the diffraction patterns of feroxyhyte are very similar (there are no extra peaks), except for the amorphous part. There is a paper https://doi.org/10.1016/j.apcatb.2012.02.026, where this halo refers to ferrihydrite. Very little information is available about ferrihydrite, especially with the same radiation source. According to Mössbauer spectroscopy, "the calculated hyperfine MS parameters can correspond to akaganeite - β-FeOOH [doi: 10.1023/A:1011076308501, doi: 10.1016/S1003-6326(21)65721-7], and ordered ferrihydrite [doi: 10.3390/agronomy10121891 , doi: 10.3389/fpls.2019.00413].". Probably β-FeOOH contributes, but the ferrihydrite phase predominates.
Kim, B. G., Park, J., Choi, W., Han, D. S., Kim, J., & Park, H. (2021). Electrocatalytic arsenite oxidation using iron oxyhydroxide polymorphs (α-, β-, and γ-FeOOH) in aqueous bicarbonate solution. Applied Catalysis B: Environmental, 283, 119608. doi:10.1016/j.apcatb.2020.119608 X-ray diffractometer (XRD, Max-2500 V, Rigaku) with Cu-Kα radiation
|
|||||||||||||||||||||||||||||||||||||||||||||||||||||||||||||||||||||
|
27. |
Line 176-177: "Table 1 presents the lattice parameters of each sample and the for each refinement: Goodness of Fit (GOF), which is the RWP/Rexp ratio [27]." better to be "Table 1 presents the lattice parameters of each sample and the Goodness of Fit (GOF), which is the RWP/Rexp ratio [27]."
|
Thanks, done |
|||||||||||||||||||||||||||||||||||||||||||||||||||||||||||||||||||||
|
28. |
There is no GOF parameter in the Table 1!
|
A related statistical concept is that of “Chi squared” orχ2 . This can be thought about by again considering that the expected value for (yC,i− yO,i )2 /Ϭ2 [yO,i] will be one, when the model is ideal and s.u. values are correct. The χ2 term is then defined as the average of these values χ2=(1/N)i(yC,i− yO,i )2 /Ϭ2 [yO,i ] Young, 1993. Note that χ 2 can also be determined from the expected and weighted profile R factors χ2=(Rwp /Rexp)2 . The single-crystal literature often uses the term goodness of fit (G) which is defined by G2= χ2 . Goodness of fit is less commonly used in powder diffraction. For reasons unclear to me, one never sees a reference to χ, only χ2 . It should be noted that χ2 should never drop below one, or equivalently, the smallest that Rwp should ever be is Rexp (see DOI: 10.1154/1.2179804) |
|||||||||||||||||||||||||||||||||||||||||||||||||||||||||||||||||||||
|
29. |
Table 1: what does it mean "Structure Fe2.66O4 Fe2.56O4"? How the numbers 2.66 and 2.56 were obtained? Please put in the manuscript.
|
The formula for non-stoichiometric magnetite Fe3-δO4 was calculated using the method proposed by Gorski [Gorski C.A. Scherer M.M. Determination of nanoparticulate magnetite stoichiometry by Mossbauer spectroscopy, acid dissolution, and powder X-ray diffraction: A critical review. Am. Mineral. 2010, 95(7), 1017–1026. DOI: 10.2138/am.2010.3435]. Magnetite can have a range of oxidation states dependent upon the amount of structural Fe2+, which can be discussed quantitatively as the magnetite stoichiometry (x = Fe2+/Fe3+). For magnetite with an ideal Fe2+ content (assuming the Fe3O4 formula), the mineral phase is known as stoichiometric magnetite (x = 0.50). As magnetite becomes oxidized, the Fe2+/Fe3+ ratio decreases (x<0.50), with this form denoted as nonstoichiometric or partially oxidized magnetite. When the magnetite is completely oxidized (x = 0), the mineral is known as maghemite (γ-Fe2O3). For nonstoichiometric magnetite, the structure is often written as Fe3–δO4, where δ can range from zero (stoichiometric magnetite) to 1/3 (completely oxidized). The stoichiometry can easily be converted to and from this form by the following relationship:
|
|||||||||||||||||||||||||||||||||||||||||||||||||||||||||||||||||||||
|
30. |
The Fe2.56O4 must have Fe(+4).
|
Obviously, the reviewer mistyped and meant "Fe2.66O4", then 2.66*(+3) = 7.98 ≈ 4*2, taking into account the error in determining the stoichiometry 0.01*3=0.03.
|
|||||||||||||||||||||||||||||||||||||||||||||||||||||||||||||||||||||
|
31. |
Why magnetite and gamma-Fe2O3 have certain composition like Fe2.66O4 and the other nanomaterials are not? Please put in all.
|
This method of calculation of stoichiometry is applicable only for magnetite, implying a priori that it is a mixture of magnetite and maghemite.
|
|||||||||||||||||||||||||||||||||||||||||||||||||||||||||||||||||||||
|
32. |
Why Fe3O4 composition is so far from Fe3O4 and is Fe2.66O4?
|
Probably, the partial oxidation of magnetite to maghemite (Fe2.66O4 instead Fe3O4) is associated with the formulation in air and insufficiently alkaline pH. In this case, the formation of a mixture of magnetite and maghemite Fe2.56O4 instead Fe2O3, on the one hand, can also be associated with insufficient oxidation of the Fe2.66O4 during synthesis (see formulation 2.3). On the other hand, Mössbauer spectroscopy is a more sensitive method, and according to it, magnetite and maghemite are the same structure in this case. |
|||||||||||||||||||||||||||||||||||||||||||||||||||||||||||||||||||||
|
33. |
What is the method for "Composition determined"? Please put in the manuscript.
|
We have done (lines 191-201) |
|||||||||||||||||||||||||||||||||||||||||||||||||||||||||||||||||||||
|
34. |
Why uncertainty of crystallite size have only Fe3O4 and gamma-Fe2O3, but the others have not?
|
Added
|
|||||||||||||||||||||||||||||||||||||||||||||||||||||||||||||||||||||
|
35. |
Lines 184-198: the Fe3O4 in fact is gamma-Fe2O3.
|
Yes, Mössbauer spectroscopy showed gamma-Fe2O3. |
|||||||||||||||||||||||||||||||||||||||||||||||||||||||||||||||||||||
|
36. |
Further discussion of titration results should keep it in focus.
|
Done |
|||||||||||||||||||||||||||||||||||||||||||||||||||||||||||||||||||||
|
37. |
Line 224: "akaganite" to replace with "akaganeite".
|
Thanks, done |
|||||||||||||||||||||||||||||||||||||||||||||||||||||||||||||||||||||
|
38. |
Table 3: please check the coercive power for ferrihydrite and goetite.
|
We have corrected magnetic characteristics, namely coercive power
|
|||||||||||||||||||||||||||||||||||||||||||||||||||||||||||||||||||||
|
39. |
Line 259-560: "the presence of a steep rise in the isotherm at low pressures may indicate the presence of micropores". That feature mentioned for alfa-FeOOH, but it is the smallest one among the others. Please reformulate to underline larger micropore area in the other samples.
|
Adsorption/desorption isotherms of nitrogen for iron nanoparticles are presented on the Figure 4. All samples are characterized by type IV isotherms according to the classi-fication, which indicates the occurrence of polymolecular adsorption and the presence of capillary condensation in mesopores. Samples of Fe3O4, γ-Fe2O3, δ-FeOOH have an H1-type hysteresis loop associated with the filling of mesopores due to capillary con-densation. The α-FeOOH sample has a hysteresis loop of the H4 type. 5Fe2O3·9H2O has a hysteresis loop of the H2(a) type with a corpuscular struc-ture, but the distribution and shape of the pores are inhomogeneous in this case. The presence of a steep rise in the isotherms at low pressures may indicate the presence of micropores. |
|||||||||||||||||||||||||||||||||||||||||||||||||||||||||||||||||||||
|
40. |
Line 286: the Supplementary materials doesn't contain Table S2. Please put it or change to Fig. S2. |
Corrected |
|||||||||||||||||||||||||||||||||||||||||||||||||||||||||||||||||||||
|
41. |
Line 287: "As the results of complexometric reactions showed (Figure 5)". The Fig. 5 do not contain reactions. Better to use "According to complexometric results (Fig. 5)".
|
Corrected, thanks |
|||||||||||||||||||||||||||||||||||||||||||||||||||||||||||||||||||||
|
42. |
Line 292: "316 mg/L for Fe2+ " but it is only 236 mg/L according to Fig. 5. Where are correct data?
|
316 mg/L is the correct data. We have changed the size of the axis so that the point gets on the figure. |
|||||||||||||||||||||||||||||||||||||||||||||||||||||||||||||||||||||
|
43. |
Line 293, 373: "can lead to uncontrolled catalysis way". Please explain what do you mean? Larger catalyst concentration in a cancer cell may produce more than 100% of ROS from same amount of H2O2? Isn't a hydrogen peroxide indeed a limitation factor for more ROS production in a cell?
|
According to the Fenton reaction, both iron ions and hydrogen peroxide together are limiting factors. Both an increase in the concentration of iron ions and an increase in the volume of hydrogen peroxide can lead to an increase in the concentration of ROS. By "uncontrolled catalysis" we mean the rapid release of iron ions in the first hour, while magnetite, like maghemite, releases in a sustained manner, approximately the same concentration of ions over 24 hours. |
|||||||||||||||||||||||||||||||||||||||||||||||||||||||||||||||||||||
|
44. |
Moreover, looking on fast decay of Fe concentration isn't it better to use long lasting effect of alfa-FeOOH ruther than the others?
|
On the one hand, you are right, alfa-FeOOH releases more iron ions, which is of course most effective in initiating the Fenton reaction. However, the release can hardly be called prolonged, since there is a sharp increase in the concentration of ions for 1-3, and then a decrease. On the other hand, magnetite and maghemite release less iron ions, but more evenly, with a prolonged release. Probably, the selection of the optimal drug still requires further evaluation. alfa-FeOOH can be utilized as aid therapy agent. |
|||||||||||||||||||||||||||||||||||||||||||||||||||||||||||||||||||||
|
45. |
Lines 295-297: "the concentration of ions in the solution increases for 3 hours. After 24 hours a decrease in the concentration of ions is observed, which may indicate the adsorption of ions from the solution onto the NPs surface". So what was originally the driving force of Fe ions desorption for the first 3 hours? Following your idea, after 3 hours, Fe ions realize that they take a wrong decision to desorb from surface?
|
Dr. Prof. Scherer in various publications describes the sorption-desorption of iron ions in relation to iron oxides and oxyhydroxides. In particular, they described the mechanisms of Fe(II) sorption on the surface of Goethite and atom exchange between goethite and aqueous Fe(II) [Handler, R. M., Beard, B. L., Johnson, C. M., & Scherer, M. M. (2009). Atom Exchange between Aqueous Fe(II) and Goethite: An Fe Isotope Tracer Study. Environmental Science & Technology, 43(4), 1102–1107. doi:10.1021/es802402m]. The authors showed complex mechanism of interaction between ions in solution and iron oxides and oxyhydroxides showing that “sorption of Fe(II) onto goethite results in electron transfer between the sorbed Fe(II) and the structural Fe(III) in goethite”. The approach used in this study is similar to that followed by Skulan et al. (27), Poulson et al. (28), Welch et al. (29), and Shahar et al. (30) to identify isotope exchange between aqueous Fe(III) and hematite, aqueous Fe(III) and ferrihydrite, aqueous Fe(III) and Fe(II), and magnetite and fayalite, respectively. Although sorption of Fe(II) to goethite and hematite occurs over a wide range of pH values (5, 16), and interfacial electron transfer has been demonstrated on these oxides (10, 17), no phase transformations have been observed, and little to no reductive dissolution of Fe(II) occurs after reaction of Fe(II) with goethite or hematite, respectively (12). Pedersen et al. incorporated 55Fe into several different Fe oxides and observed the release of 55Fe into solution upon exposure to aqueous Fe(II) (12). Release of 55Fe into solution was observed for lepidocrocite, ferrihydrite, and goethite, but not for hematite. On the other side, the observed decrease in the concentration of ions within 24 hours may be due not to the sorption of ions, but to the formation of insoluble hydroxocomplexes Fe(OH)3 and Fe(OH)2+, which are not detected by the selected complexometric methods. |
|||||||||||||||||||||||||||||||||||||||||||||||||||||||||||||||||||||
|
46. |
Lines 298-318: since there are shell formation in composite (or micelle), these references are hard to compare to discussed results. What shell on maggemite NP do you expect in your compexation experiment?
|
This part was deleted due to incorrect comparison |
|||||||||||||||||||||||||||||||||||||||||||||||||||||||||||||||||||||
|
47. |
Lines 319-320: "practically insoluble Fe(OH)3 complexes from Fe3+ ions under these conditions." But the Fe3+ ions you mention in that sentence come from what source? From much more soluble NP? The "Fe(OH)3 complexes" are molecular uncharged species? Than why it is insoluble? The Si(OH)4 molecule for instance, is soluble. Such a species are highly reactive and may react with ligands like SCN-. Please give the references to approve your point.
|
Given the diluted solution of Fe-based NPs, Fe3+ come from NPs surface. According to Fe3+ concentration as 180 mg/L and pH of solution as 6-8, Fe(OH)3 species are dominant. Pls see calculation of concentration of hydroxo complexes and their distribution at various pH: The Fe3+ are hydrolyzed in dilute solutions in neutral and alkaline media and resulted in hydroxo complexes. To calculate the active form of the existence of Fe3+ ions at different pH in aqueous solutions, the following equilibria were compiled for the forms of Fe3+ ions and reference data of hydrolysis constants were used (Battler, Ionic equilibriums).
The equilibrium equations for hydroxoforms of iron (III) ions are presented below: where К1, К2, К3, К4 - stepwise equilibrium constants for the formation of iron(III) hydroxocomplexes in solution. The balance equation is as follows: :
СFe ≈ [Fe3+]+[FeOH2+]+[Fe(OH)2+]+[Fe(OH)3]+[Fe(OH)4-]
Concentrations of various hydroxocomplexes of Fe3+ ions in 0.001 M solution (-lg[Fe(III)])
Distribution of various hydroxocomplexes of iron (III) in 0.001 M solution in the composition of pH 2÷10 (molar %)
|
|||||||||||||||||||||||||||||||||||||||||||||||||||||||||||||||||||||
|
48. |
Line 322: reformulate "release of α-FeOOH ions".
|
Thanks, done |
|||||||||||||||||||||||||||||||||||||||||||||||||||||||||||||||||||||
|
49. |
Line 326: "their microstructure, shown by Mössbauer spectroscopy and XRD". The therm "microstructure" looks unsuitable. XRD provide structural parameters, Mössbauer spectroscopy provide local surrounding data for Fe atoms/ions. Please select proper therm.
|
We have corrected, thanks |
|||||||||||||||||||||||||||||||||||||||||||||||||||||||||||||||||||||
|
50. |
Line 328: "at a pH close to neutral (6.6 for γ-Fe2O3 and 7.0 for Fe3O4)". Line 350: "the pH of the solution, which was 3.45" So what was the pH of the solutions you used for testing of NP?
|
For a preliminary assessment of the ability of the selected preparations to release iron ions, all preparations were studied with native pH Fe3O4 pH 7γ-Fe2O3 pH 6.6 α-FeOOH pH 6.9 δ-FeOOH pH 6.7 5Fe2O3·9H2O pH 6.9 We have added the pH values to the article. |
|||||||||||||||||||||||||||||||||||||||||||||||||||||||||||||||||||||
|
51. |
Please put description of compleximetric experiment (see in the beginning lines 158-160): please indicate components and concentrations. Line 159-160: "in buffer solutions containing potassium thiocyanate (50% solution) and HCl (18.25% solution)... and potassium hexacyanoferrate(III) (1% solution)". What is the aim of buffer solution if next to that you adding 18% HCl? Please put the reason for HCl addition and volumes of KSCN and HCl solutions. Please indicate the target pH upon adding HCl.
|
To detect iron ions, powder were suspended in deinized water, then subjected to centrifugation (6000 rpm, 5 min) after 0, 0.5, 1, 3, and 24 hours. Appropriate ion detection reagents were added to the supernatant after separation. To detect Fe3+ ions, 200 μl of potassium thiocyanate KSCN (50% solution) and 200 μl of HCl (18.25% solution) were added to 5 ml of supernatant, since the reaction between the Fe3+ ions and KSCN proceeds in a strongly acidic medium at pH close to 2. Then the solution was kept for 20 minutes to reach equilibrium and the absorption spectrum in the region of 490 nm was measured. To detect Fe2+ ions, 2 ml phenanthroline C12H8N2∙H2O (2.5% solution) and 600 μl of ammonium acetate buffer solution (250 ml of NH4OH and 900 ml of glacial acetic acid) were also added to the 5 ml of supernatant. Further, the solution was also kept for 20 minutes to reach equilibrium (no color change) and examined in the region of 690 nm. The absorbance was detected by UV–Vis-NIR spectrophotometry (Cary UV-Vis-NIR Spectrophotometer, Agilent Technologies)
|
|||||||||||||||||||||||||||||||||||||||||||||||||||||||||||||||||||||
|
52. |
Importance of that you underline on lines 298-300 etc.
|
deleted |
|||||||||||||||||||||||||||||||||||||||||||||||||||||||||||||||||||||
|
53. |
"At different time points (0, 0.5, 1, 3 and 24 h), the mixtures were centrifuged" - does it mean, that oxyhydrate nanoparticles, buffer, KSCN, HCl, K3[Fe(CN)6] all were in single flask? |
According to the procedure, sample suspensions were stored for 0, 0.5, 1, 3, and 24 hours; then, the supernatant was separated by centrifugation. Reagents for the determination of iron ions (method line 159-170) were already added to the supernant. Solutions with reagents were examined 20 minutes after preparation. |
|||||||||||||||||||||||||||||||||||||||||||||||||||||||||||||||||||||
|
54. |
Line 328: "Fe2+ ... are rapidly oxidized to Fe3+ ". Please explain why it is not oxidized in case of delta-FeOOH? Isn't Fe2+ more rapidly reacted with 1% solution K3[Fe(CN)6] rather than with O2?
|
Fe2+ are likely more rapidly reacted with 1% solution K3[Fe(CN)6] rather than with O2.
|
|||||||||||||||||||||||||||||||||||||||||||||||||||||||||||||||||||||
|
55. |
Say, (NH4)2Fe(SO4)2 solution is stable enough for hours in air and was widely used for titrimetric analysis for 1800-1950 min period.
|
No, the solutions after addition of KSCN and HCl for detection of Fe3+ and phenanthroline and ammonium acetate buffer for Fe2+ was examined 20 minutes after preparation. |
|||||||||||||||||||||||||||||||||||||||||||||||||||||||||||||||||||||
|
56. |
See also line 340: "Fe3+/Fe2+ remains almost unchanged and close to 5 during the entire observation time", so the concentrations changes 3-4 times, but the oxidation of Fe(+2) is not detected. See also line 349. |
We apologize, but the original data (see Suppl, Fig 2) has been revised. According to dara, δ-FeOOH releases only Fe3+ ions. All parts of the discussion regarding the ratio of ions for this sample have been deleted. |
|||||||||||||||||||||||||||||||||||||||||||||||||||||||||||||||||||||
|
57. |
Line 333-334: poor English.
|
English was corrected. |
|||||||||||||||||||||||||||||||||||||||||||||||||||||||||||||||||||||
|
58. |
Line 344: the therms "adsorption and desorption of ions" is much more preferable instead of "ion release" (see line 336 and others). "Release" is well applicable to the well soluble substance, like "release of ascorbic acid from gelly capsula" or "release of HCN from apricot shell". "Release" is non-reversible process. The nonstationary process you discuss is reversible and may be connected to adsorption and desorption. |
Thank you for explanation. We cannot confidently consider this process as sorption-desorption, therefore we propose to leave the term release, since it is also used by the authors of similar research [Zohdi et al., Corrosion performance and metal ion release of amorphous and nanocrystalline Fe-based alloys under simulated body fluid conditions, Materials letters, 2013 http://dx.doi.org/10.1016/j.matlet.2012.12.051; El-Fiqi et al., Iron ions-releasing mesoporous bioactive glass ultrasmall nanoparticles designed as ferroptosis-based bone cancer nanotherapeutics: Ultrasonic coupled sol–gel synthesis, properties and iron ions release, Materials letters, 2021 https://doi.org/10.1016/j.matlet.2021.129759; He et al., Ferric ions release from iron-binding protein: Interaction between acrylamide and human serum transferrin and the underlying mechanisms of their binding, Science of the Total Environment, 2022 http://dx.doi.org/10.1016/j.scitotenv.2022.157583] |
|||||||||||||||||||||||||||||||||||||||||||||||||||||||||||||||||||||
|
59. |
Line 346: "50 times more Fe3+ and 150 times less Fe2+ than" to replace with "50 times more Fe3+ and 150 times more Fe2+ than". |
Corrected, thanks |
|||||||||||||||||||||||||||||||||||||||||||||||||||||||||||||||||||||
|
60. |
Line 354: "316 mg/L" but it is only 236 mg/L according to Fig. 5. Where are correct data?
|
316 mg/L is the correct data. We have changed the size of the axis so that the point gets on the figure. |
|||||||||||||||||||||||||||||||||||||||||||||||||||||||||||||||||||||
|
61. |
Conclusions: poor English. Please reformulate.
|
done |
|||||||||||||||||||||||||||||||||||||||||||||||||||||||||||||||||||||
|
62. |
Line 362: "mixture of magnetite and maghemite with composition as Fe2.66O4 instead of Fe3O4." I doesn't see evidence for mixture. According to Moessbauer it is maghemite. It also maghemite according to Fe(+2) complexometric study. PXRD data and magnetic measurements are fit well with maghemite. |
For the calculation, we used the approach proposed by Gorsky [Gorski C.A. Scherer M.M. Determination of nanoparticulate magnetite stoichiometry by Mossbauer spectroscopy, acid dissolution, and powder X-ray diffraction: A critical review. Am. Mineral. 2010, 95(7), 1017–1026. DOI: 10.2138/am.2010.3435], a priori assuming that nonstoichiometric magnetite is a mixture of magnetite and maghemite. According to Schwaminger et al. 2017 [DOI: 10.1039/c6ce02421a], the magnetite content calculated from data from different methods such as XRD, Mössbauer spectroscopy and Raman may differ significantly. We tried to evaluate Fe3O4 and Fe2O3 analyzing peak 440 on the XRD. Deconvolution of the peak showed that it is the sum of the peaks ~62° and ~63° both for Fe3O4 and Fe2O3, that maybe used as a proof for the presence of magnetite. Regarding the complexometric method, it may not be sensitive enough to detect Fe2+ ions. |
|||||||||||||||||||||||||||||||||||||||||||||||||||||||||||||||||||||
|
63. |
Do you have EXAFS or XANES data to approve Fe(II) in that sample?
|
No, unfortunately, we don’t have |
|||||||||||||||||||||||||||||||||||||||||||||||||||||||||||||||||||||
|
64. |
Line 372: "316 mg/L" but it is only 236 mg/L according to Fig. 5. Where are correct data?
|
316 mg/L is the correct data. We have changed the size of the axis so that the point gets on the figure. |
|||||||||||||||||||||||||||||||||||||||||||||||||||||||||||||||||||||
|
65. |
Fig. S2 is hard to read. Numbers are small, lines are of same color, lines are too close to each other. Please remove part of lines, change the line type (dot, dash-dot) and its colors and thickness to make graph easy to read.
|
We have separated the charts for Fe2+ and Fe3+, and also changed the numbers and symbols on the axes more |
|||||||||||||||||||||||||||||||||||||||||||||||||||||||||||||||||||||
|
66. |
The absorbance at Fig. S2(e) is very similar or smaller to (c) and (d). From manuscript Fig. 5 the α-FeOOH (e) is orders of magnitude larger values of Fe ions release comparing to (c) and (d). Please explain it and make it clear in the text. For instance, put mass of the samples used on Fig. 2S.
|
Due to the very high optical density of the supernatants of the original preparations (above 2, while the calibration allows the use of values up to 1.5 for a linear relationship), all supernatants of the α-FeOOH preparation were diluted 50 times.
|
|||||||||||||||||||||||||||||||||||||||||||||||||||||||||||||||||||||
|
67. |
For the Fig. S2 it is recommended to present additional graph (f) with Fe(+3)-SCN complex spectrum and separately Prussian blue spectrum.
|
The presented spectra are already the spectra of Fe(+3)-SCN and Fe(2+)-phenanthroline complexes, since it was the supernatant that was mixed with the reagents.
|

Reviewer 2 Report
The research in this paper has very important practical significance, and has a high perspective for the application of iron containing nanomaterials.
1, The description of the synthesis methods of the main materials is not clear enough, and many only give references, which is not enough.
2, The fonts in most of the Figures are too small, especially in Figures 1, 2 and 4.
3, In the same article, the full name and abbreviation of a graph should be unified. Don't sometimes use the Figure and sometimes Fig..
Author Response
Dear Reviewer,
Thanks for reading the article in detail. We have made the changes you suggested.
|
1, The description of the synthesis methods of the main materials is not clear enough, and many only give references, which is not enough. |
Lines 99-137 |
You are absolutely right. We have detailed all the syntheses, indicating the temperature and drying time, stirring speed, etc. |
|
2, The fonts in most of the Figures are too small, especially in Figures 1, 2 and 4. |
Lines 322, 407, 460 |
We have increased the size of the fonts to make them more readable. Thanks a lot. |
|
3, In the same article, the full name and abbreviation of a graph should be unified. Don't sometimes use the Figure and sometimes Fig. |
The whole paper |
We brought the names to uniformity throughout the text |

Round 2
Reviewer 1 Report
Second review
12.12.2022
Manuscript get much better and could be accepted after minor revision.
Line 29: FeOOH is not a "complex" and not an "oxide" and not a "complex oxide". The controversial results - very high Fe(2+) in a product you claim as purely Fe(3+) phase is disappointing. I would recommend to remove that compound from manuscript or lable it everywhere as alfa-FeOOH-Fe2+. Moreover, the strange procedure of synthesis of that product (using highly reducing NaBH4) authors are not referred to any literature source.
Line 76: move "
Line 86: remove ", and - due to low oxidation in acid medium that is important for controlled drug release".
Line 130 (authors miss to answer this question): the α-FeOOH is a Fe(+3) compound. The described procedure likely to produce product containing Fe(OH)2.
Please indicate the color of alfa-FeOOH you obtain.
For instance in [https://doi.org/10.1007/s11051-008-9467-z] authors uses Fe(+3) nitrate:
"Precipitation of α-FeOOH in aqueous solution was performed by placing 0.025 mol dm−3 solution of [Fe(NO3)3 · 4H2O—BDH] at pH 2 overnight in a water-bath set at 50 °C. The pH of the solution was adjusted to 2 using nitric acid. A yellow precipitate formed and separated from the acidic solution by centrifugation."
Please put the literature references to the method you used. Otherwise the product likely to be Fe(2+) or Fe(2+)/Fe(3+) hydroxide (it could be minor amorphous impurity).
Line 151: to replace "Before the start of the tests, the samples were degassed, which consisted in their heating in a He (helium) current, 150°C, 2 h, flow rate 0.2 L/h" with "Before the start of the tests, the samples were degassed by heating in a helium flow of 0.2 L/h, 150°C, 2 h".
Table 1, Line 201: Fe3O4 and gamma-Fe2O3 are both have same 2.66 in Table 1. Is it correct?
Table 3: for weakly magnetic materials (delta- and alfa-FeOOH, 5Fe2O3*9H2O) the coercive power expect to be much smaller than Fe3O4 and maghemite.
Please check again!
Fig. 5: Why the curves are not fit with approved error bar? Either curve is wrong, or the error bar is wrong. Please select.
Line 353: what is ZVI?
Line 364: "mixture of α-FeOOH with Fe2+." to replace with "Fe(2+)-doped alfa-FeOOH".
Line 365: Fe3O4 - formate it.
Line 366: Fe0 to replace with gamma-Fe2O3!

Author Response
Dear Reviewer,
Thank you for your re-review and useful comments. We made corrections to the text of the pare and table below.
|
Line 29: FeOOH is not a "complex" and not an "oxide" and not a "complex oxide". The controversial results - very high Fe(2+) in a product you claim as purely Fe(3+) phase is disappointing. I would recommend to remove that compound from manuscript or lable it everywhere as alfa-FeOOH-Fe2+. Moreover, the strange procedure of synthesis of that product (using highly reducing NaBH4) authors are not referred to any literature source. |
In this line we have replace to “compound” and marked everywhere as α-FeOOH-Fe2+ |
|
Line 76: move "
|
done |
|
Line 86: remove ", and - due to low oxidation in acid medium that is important for controlled drug release".
|
done |
|
Line 130 (authors miss to answer this question): the α-FeOOH is a Fe(+3) compound. The described procedure likely to produce product containing Fe(OH)2. Please indicate the color of alfa-FeOOH you obtain. For instance in [https://doi.org/10.1007/s11051-008-9467-z] authors uses Fe(+3) nitrate: "Precipitation of α-FeOOH in aqueous solution was performed by placing 0.025 mol dm−3 solution of [Fe(NO3)3 · 4H2O—BDH] at pH 2 overnight in a water-bath set at 50 °C. The pH of the solution was adjusted to 2 using nitric acid. A yellow precipitate formed and separated from the acidic solution by centrifugation." Please put the literature references to the method you used. Otherwise the product likely to be Fe(2+) or Fe(2+)/Fe(3+) hydroxide (it could be minor amorphous impurity).
|
The described procedure for Fe2+-doped alpha-FeOOH synthesis in our paper was adopted from (Lowry, G. V., & Johnson, K. M. (2004). Congener-Specific Dechlorination of Dissolved PCBs by Microscale and Nanoscale Zerovalent Iron in a Water/Methanol Solution. Environmental Science & Technology, 38(19), 5208–5216. doi:10.1021/es049835q) for nano zero valent iron (ZVI) and modified using ambient conditions instead of argon and EtOH instead of MeOH in order (i) to prevent rapid oxidation to ferrihydrite and (ii) to dope final product with Fe2+. A yellow precipitate of final product was formed.
|
|
Line 151: to replace "Before the start of the tests, the samples were degassed, which consisted in their heating in a He (helium) current, 150°C, 2 h, flow rate 0.2 L/h" with "Before the start of the tests, the samples were degassed by heating in a helium flow of 0.2 L/h, 150°C, 2 h".
|
done |
|
Table 1, Line 201: Fe3O4 and gamma-Fe2O3 are both have same 2.66 in Table 1. Is it correct?
|
We have changed on Fe2.66O4 for Fe3O4 and on Fe2.56O4 for Fe2O3 |
|
Table 3: for weakly magnetic materials (delta- and alfa-FeOOH, 5Fe2O3*9H2O) the coercive power expect to be much smaller than Fe3O4 and maghemite. Please check again!
|
The high values of the coercive force of the samples δ-FeOOH , 5Fe2O3•9H2O and α-FeOOH compared to Fe3O4 and γ-Fe2O3 can be associated with an asymmetric crystal lattice, as shown by XRD, which also increases the magnetic exchange anisotropy. According to Mössbauer spectroscopy data, the 5Fe2O3•9H2O sample can contain the β-FeOOH phase, and the α-FeOOH sample contains hydrated iron ions (+2) and iron ions (+3) in an octahedral oxygen environment, which also increases the magnetic exchange anisotropy.
|
|
Fig. 5: Why the curves are not fit with approved error bar? Either curve is wrong, or the error bar is wrong. Please select.
|
For describing of points we have used b-splain curve in Origin which shows only trend of points. |
|
Line 353: what is ZVI?
|
ZVI - zero valent iron It was replaced on Fe2+-doped α-FeOOH |
|
Line 364: "mixture of α-FeOOH with Fe2+." to replace with "Fe(2+)-doped alfa-FeOOH".
|
done |
|
Line 365: Fe3O4 - formate it.
|
done |
|
Line 366: Fe0 to replace with gamma-Fe2O3! |
done |
